# On the Convergence Rates of Federated Q-Learning across Heterogeneous Environments

## Abstract

Large-scale multi-agent systems are often deployed across wide geographic areas, where agents interact with heterogeneous environments. There is an emerging interest in understanding the role of heterogeneity in the performance of the federated versions of classic reinforcement learning algorithms. In this paper, we study synchronous federated Q-learning, which aims to learn an optimal Q-function by having $K$ agents average their local Q-estimates per $E$ iterations. We observe an interesting phenomenon on the convergence speeds in terms of $K$ and $E$. Similar to the homogeneous environment settings, there is a linear speed-up concerning $K$ in reducing the errors that arise from sampling randomness. Yet, in sharp contrast to the homogeneous settings, $E > 1$ leads to significant performance degradation. Specifically, we provide a fine-grained characterization of the error evolution in the presence of environmental heterogeneity, which decay to zero as the number of iterations $T$ increases. The slow convergence of having $E > 1$ turns out to be fundamental rather than an artifact of our analysis. We prove that, for a wide range of stepsizes, the $\ell_\infty$ norm of the error cannot decay faster than $\Theta(E/T)$. In addition, our experiments demonstrate that the convergence exhibits an interesting two-phase phenomenon. For any given stepsize, there is a sharp phase-transition of the convergence: the error decays rapidly in the beginning yet later bounces up and stabilizes. Provided that the phase-transition time can be estimated, choosing different stepsizes for the two phases leads to faster overall convergence.

## 1 Introduction

Advancements in unmanned capabilities are rapidly transforming industries and national security by enabling fast-paced and versatile operations across domains such as advanced manufacturing (Park et al., 2019), autonomous driving (Kiran et al., 2021), and battlefields (Möhlenhof et al., 2021). Reinforcement learning (RL) – a cornerstone for unmanned capabilities – is a powerful machine learning method that aims to enable an agent to learn an optimal policy via interacting with its operating environment to solve sequential decision-making problems (Bertsekas & Tsitsiklis, 1996; Bertsekas, 2019). However, the ever-increasing complexity of the environment results in a high-dimensional state-action space, often imposing overwhelmingly high sample collection requirements on individual agents. This limited-data challenge becomes a significant hurdle that must be addressed to realize the potential of reinforcement learning.

In this paper, we study reinforcement learning within a federated learning framework (also known as Federated Reinforcement Learning (Qi et al., 2021; Jin et al., 2022; Woo et al., 2023)), wherein multiple agents independently collect samples and collaboratively train a common policy under the orchestration of a parameter server without disclosing the local data trajectories. A simple illustration can be found in Fig. 1. When the environments of all agents are homogeneous, it has been shown that the federated version of classic reinforcement learning algorithms can significantly alleviate the data collection burden on individual agents (Woo et al., 2023; Khodadadian et al., 2022) – error bounds derived therein exhibit a linear speedup in terms of number of agents.

Moreover, by tuning the synchronization period $E$ (i.e., the number of iterations between agent synchronization), the communication cost can be significantly reduced compared with $E = 1$ yet without significant

performance degradation. However, many large-scale multi-agent systems are often deployed across wide geographic areas, resulting in agents interacting with heterogeneous environments. For instance, connected and autonomous vehicles (CAVs) operating in various regions of a metropolitan area encounter diverse conditions such as varying traffic patterns, road infrastructure, and local regulations. The clients' federation must be managed in a way that ensures the learned policy is robust to environmental heterogeneity.

There is an emerging interest in mathematically understanding the role of heterogeneity in the performance of the federated versions of classic reinforcement learning algorithms (Jin et al., 2022; Woo et al., 2023; Doan et al., 2019; Wang et al., 2023; Xie & Song, 2023) such as Q-learning, policy gradient methods, and temporal difference (TD) methods. In this paper, we study synchronous federated Q-learning in the presence of environmental heterogeneity, which aims to learn an optimal Q-function by averaging local Q-estimates per $E$ (where $E \geq 1$) update iterations on their local data. We leave the exploration of asynchronous Q-learning for future work. Federated

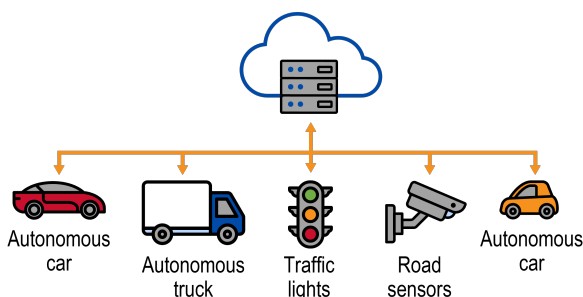

Figure 1: An illustration of a federated learning system.

Q-learning is a natural integration of FedAvg and Q-learning (Jin et al., 2022; Woo et al., 2023). The former is the most widely adopted classic federated learning algorithm (Kairouz et al., 2021; McMahan et al., 2017), and the latter is one of the most fundamental model-free reinforcement learning algorithms (Watkins & Dayan, 1992). Despite intensive study, the tight sample complexity of Q-learning in the single-agent setting was open until recently (Li et al., 2024). Similarly, the understanding of FedAvg is far from complete; a detailed discussion can be found in Section 2.

**Contributions.** In this paper, we study synchronous federated Q-learning in the presence of environment heterogeneity.

- We provide a fine-grained characterization of the error evolution, which decays to zero as the number of iterations $T$ increases. We observe an interesting phenomenon on the convergence speeds in terms of $K$ and $E$. Similar to the homogeneous environment settings, there is a linear speed-up concerning $K$ in reducing the errors that arise from sampling randomness. Yet, in sharp contrast to the homogeneous settings, $E > 1$ leads to significant performance degradation.

- We prove that the convergence slowing down for $E > 1$ is fundamental. We show that the $\ell_\infty$ norm of the error cannot decay faster than $\Theta(E/T)$. A practical implication of this impossibility result is that, eventually, having multiple local updates (i.e., $E > 1$) ends up consuming more samples (i.e., $E \times$ more) than using $E = 1$.

- Our numerical results illustrate that when the environments are heterogeneous and $E > 1$, and there exists a sharp phase-transition of the error convergence: The error decays rapidly in the beginning yet later bounces up and stabilizes. In addition, provided that the phase-transition time can be estimated, choosing different stepsizes for the two phases can lead to faster overall convergence.

## 2 Related Work

**Federated Learning.** Federated learning is a communication-efficient distributed machine learning approach that enables training global models without sharing raw local data (McMahan et al., 2017; Kairouz et al., 2021). Federated learning has been adopted in commercial applications that involve diverse edge devices such as autonomous vehicles (Du et al., 2020; Chen et al., 2021; Zeng et al., 2022; Posner et al., 2021; Peng et al., 2023), internet of things (Nguyen et al., 2019; Yu et al., 2020), industrial automation (Liu et al., 2020), healthcare (Yan et al., 2021; Sheller et al., 2019), and natural language processing (Yang et al., 2018; Ramaswamy et al., 2019). Multiple open-source frameworks and libraries are available such as FATE, Flower, OpenMinded-PySyft, OpenFL, TensorFlow Federated, and NVIDIA Clara.

FedAvg was proposed in the seminal work (McMahan et al., 2017), and has been one of the most widely implemented federated learning algorithms. It also has inspired many follow-up algorithms such as FedProx (Li et al., 2020b), FedNova (Wang et al., 2020), SCAFFOLD (Karimireddy et al., 2020), and adaptive federated methods (Deng et al., 2020). Despite intensive efforts, the theoretical understanding of FedAvg is far from complete. Most existing theoretical work on FedAvg overlooks the underlying data statistics at the agents, which often leads to misalignment of the pessimistic theoretical predictions and empirical success (Su et al., 2023; Pathak & Wainwright, 2020; Wang et al., 2022a;b). This theory and practice gap is studied in a recent work (Su et al., 2023) in the context of solving general non-parametric regression problems. It shows that the limiting points of the global model under FedAvg is one unbiased estimator of the underlying model that generates the data.

**Reinforcement Learning.** There has been extensive research on the convergence guarantees of reinforcement learning algorithms. A recent surge of work focuses on non-asymptotic convergence and the corresponding sample complexity for the single-agent setup. Bhandari et al. (2018) analyses non-asymptotic TD learning with linear function approximation (LFA) considering a variety of noise conditions, including noiseless, independent noise and Markovian noise. The results were extended to TD($\lambda$) and Q-learning. Li et al. (2020a) investigates the sample complexity of asynchronous Q-learning with different families of learning rates. They also provide an extension of using variance reduction methods inspired by the seminal SVRG algorithm. Li et al. (2024) shows the sample complexity of Q-learning. Let $\mathcal{A}$ be the set of actions. When $|\mathcal{A}| = 1$, the sample complexity of synchronous Q-learning is sharp and minimax optimal, however, when $|\mathcal{A}| \geq 2$, it is shown that synchronous Q- learning has a lower bound which is not minimax optimal.

**Federated Reinforcement Learning.** Woo et al. (2023) provides sample complexity guarantees for both synchronous and asynchronous distributed Q-learning and reveals that given the same transition probability (i.e., homogeneous environment) for all agents, they can speed up the convergence process linearly by collaboratively learning the optimal Q-function. Doan et al. (2019) investigates distributed Temporal Difference (TD) algorithm TD(0) with LFA under the setting of multi-agent MDP, where multiple agents act in a shared environment and each agent has its own reward function. They provide a finite-time analysis of this algorithm that with constant stepsize, the estimates of agents can converge to a neighborhood around optimal solutions at the rate of $\mathcal{O}(1/T)$ and asymptotically converge to the optimal solutions at the rate of $\mathcal{O}(1/\sqrt{T+1})$, where $T$ is the timestep. Khodadadian et al. (2022) studies on-policy federated TD learning, off-policy federated TD learning, and federated Q-learning of homogeneous environment and reward with Markovian noise. The sample complexity derived exhibits linear speedup with respect to the number of agents. Heterogeneous environments are considered in Jin et al. (2022); Wang et al. (2023); Xie & Song (2023); Zhang et al. (2023b). Jin et al. (2022) studies federated Q-learning and policy gradient methods under the setting of different known transition probabilities for each agent. Yet, no state sampling is considered. Wang et al. (2023) proposes FedTD(0) with LFA dealing with the environmental and reward heterogeneity of MDPs. They rigorously prove that in a low-heterogeneity regime, there is a linear convergence speedup in the number of agents. Xie & Song (2023) uses KL-divergence to penalize the deviation of local update from the global policy, and they prove that under the setting of heterogeneous environments, the local update is beneficial for global convergence using their method. Zhang et al. (2023a) proposes FedSARSA using the classic on-policy RL algorithm SARSA with linear function approximation (LFA) under the setting of heterogeneous environments and rewards. They theoretically prove that the algorithm can converge to the near-optimal solution. Neither Xie & Song (2023) nor Zhang et al. (2023a) characterize sample complexity.

# 3 Preliminary on Q-Learning

**Markov decision process.** A Markov decision process (MDP) is defined by the tuple $\langle \mathcal{S}, \mathcal{A}, \mathcal{P}, \gamma, R \rangle$, where $\mathcal{S}$ represents the set of states, $\mathcal{A}$ represents the set of actions, the transition probability $\mathcal{P} : \mathcal{S} \times \mathcal{A} \to [0,1]$ provides the probability distribution over the next states given a current state $s$ and action $a$, the reward function $R : \mathcal{S} \times \mathcal{A} \to [0,1]$ assigns a reward value to each state-action pair, and the discount factor $\gamma \in (0,1)$ models the preference for immediate rewards over future rewards. It is worth noting that $\mathcal{P} = \{P(\cdot \mid s, a)\}_{s \in \mathcal{S}, a \in \mathcal{A}}$ is a collection of $|\mathcal{S}| \times |\mathcal{A}|$ probability distributions over $\mathcal{S}$, one for each state-action pair $(s, a)$.

**Policy, value function, Q-Function, and optimality.** A policy $\pi$ specifies the action-selection strategy and is defined by the mapping $\pi : \mathcal{S} \to \Delta(A)$, where $\pi(a \mid s)$ denotes the probability of choosing action $a$

when in state $s$. For a given policy $\pi$, the value function $V^\pi : \mathcal{S} \to \mathbb{R}$ measures the expected total discounted reward starting from state $s$:

$$V^\pi(s) = \mathbb{E}_{a_t \sim \pi(\cdot|s_t), s_{t+1} \sim P(\cdot|s_t, a_t)} \left[ \sum_t \gamma^t R(s_t, a_t) \mid s_0 = s \right], \quad \forall s \in \mathcal{S}.$$

The state-action value function, or Q-function $Q^\pi : \mathcal{S} \times \mathcal{A} \to \mathbb{R}$, evaluates the expected total discounted reward from taking action $a$ in state $s$ and then following policy $\pi$:

$$Q^\pi(s, a) = R(s, a) + \mathbb{E}_{a_t \sim \pi(\cdot|s_t), s_{t+1} \sim P(\cdot|s_t, a_t)} \left[ \sum_t \gamma^t \mathcal{R}(s_t, a_t) \mid s_0 = s, a_0 = a \right], \forall (s, a) \in \mathcal{S} \times \mathcal{A}.$$

An optimal policy $\pi^*$ is one that maximizes the value function for every state, that is $\forall s \in \mathcal{S}, V^{\pi^*}(s) \geq V^\pi(s)$ for any other $\pi \neq \pi^*$. Such a policy ensures the highest possible cumulative reward. The optimal value function $V^*$ (shorthand for $V^{\pi^*}$) and the optimal Q-function $Q^*$ (shorthand for $Q^{\pi^*}$) are defined under the optimal policy $\pi^*$.

The Bellman optimality equation for the value function and state-value function are:

$$V^*(s) = \max_a [R(s, a) + \gamma \sum_{s' \in \mathcal{S}} P(s'|s, a) V^*(s')]$$

$$Q^*(s, a) = R(s, a) + \gamma \sum_{s' \in \mathcal{S}} P(s'|s, a) \max_{a' \in \mathcal{A}} Q^*(s', a').$$

**Q-learning.** Q-learning (Watkins & Dayan, 1992) is a model-free reinforcement learning algorithm that aims to learn the value of actions of all states by updating Q-values through iterative exploration of the environment, ultimately converging to the optimal state-action function. Based on the Bellman optimality equation for the state-action function, the update rule for Q-Learning is formulated as:

$$Q_{t+1}(s, a) = (1 - \lambda) Q_t(s, a) + \lambda [R(s, a) + \gamma \max_{a' \in \mathcal{A}} Q_t(s', a')], \quad \forall (s, a) \in \mathcal{S} \times \mathcal{A},$$

where $s'$ is sampled from the environment or the transition probability and $\lambda$ is the stepsize.

## 4 Federated Q-learning

The federated learning system consists of one parameter server (PS) and $K$ agents. The $K$ agents are deployed in possibly heterogeneous yet independent environments. The $K$ agents are modeled as Markov Decision Processes (MDPs) with $\mathcal{M}_k = \langle \mathcal{S}, \mathcal{A}, \mathcal{P}^k, \gamma, R \rangle$ for $k = 1, \cdots, K$, where $\mathcal{P}^k = \{P^k(\cdot \mid s, a)\}_{s \in \mathcal{S}, a \in \mathcal{A}}$ are the collection of probability distributions that can be heterogeneous across agents. In the synchronous setting, each agent $k$ has access to a generative model, and generates a new state sample for each $(s, a)$ via

$$s_t^k(s, a) \sim P^k(\cdot \mid s, a)$$

i.e., $\mathbb{P}\left\{s_t^k(s, a) = s'\right\} = P^k(s' \mid s, a)$ for all $s' \in \mathcal{S}$, independently across state-action pairs $(s, a)$. For each $(s, a)$, the global environment $\bar{P}(\cdot \mid s, a)$ (Jin et al., 2022) is defined as

$$\bar{P}(s' \mid s, a) = \frac{1}{K} \sum_{k=1}^K P^k(s' \mid s, a), \forall s'. \tag{1}$$

with the corresponding global MDP defined as $\mathcal{M}_g = \langle \mathcal{S}, \mathcal{A}, \bar{\mathcal{P}}, \gamma, R \rangle$. Define transition heterogeneity $\kappa$ as

$$\sup_{k, s, a} \left\| \bar{P}(\cdot \mid s, a) - P^k(\cdot \mid s, a) \right\|_\infty := \kappa. \tag{2}$$

Let $Q^*$ denote the optimal Q-function of the global MDP. By the Bellman optimality equation, we have for all $(s, a)$,

$$Q^*(s, a) = R(s, a) + \gamma \sum_{s' \in \mathcal{S}} \bar{P}(s' \mid s, a) V^*(s'), \tag{3}$$

where $V^*(s) = \max_{a \in \mathcal{A}} Q^*(s, a)$ is the optimal value function.

The goal of the federated Q-learning is to have the $K$ agents collaboratively learn $Q^*$. We consider synchronous federated Q-learning, which is a natural integration of FedAvg and Q-learning (Woo et al., 2023; Jin et al., 2022) – described in Algorithm 1. Every agent initializes its local $Q^k$ estimate as $Q_0$ and performs standard synchronous Q-learning based on the locally collected samples $s_t^k(s,a)$. Whenever $t+1 \mod E = 0$, through the parameter server, the $K$ agents average their local estimate of $Q$; that is, all agents report their $Q_{t+\frac{1}{2}}^k$ to the parameter server, which computes the average and sends back to agents.

## 5 Main Results

With a little abuse of notation, let the matrix $P^k \in \mathbb{R}^{|\mathcal{S}||\mathcal{A}| \times |\mathcal{S}|}$ represent the transition kernel of the MDP of agent $k$ with the $(s,a)$-th row being $P^k(\cdot \mid s,a) \in \mathbb{R}^{|\mathcal{S}|}$ – the transition probability of the state-action pair $(s,a)$. For ease of exposition, we write $P^k(\cdot \mid s,a) = P^k(s,a)$ as the state transition probability at the state-action pair $(s,a)$ when its meaning is clear from the context.

---

**Algorithm 1** Synchronous Federated Q-Learning

**Inputs:** discount factor $\gamma$, $E$, total iteration $T$, stepsize $\lambda$, initial estimate $Q_0$

1: **for** $k \in [K]$ **do**
2: $\quad Q_0^k = Q_0$
3: **end for**
4: **for** $t = 0$ to $T-1$ **do**
5: $\quad$ **for** $k \in [K]$ and $(s,a) \in \mathcal{S} \times \mathcal{A}$ **do**
6: $\quad\quad Q_{t+\frac{1}{2}}^k(s,a) \quad = \quad (1-\lambda)Q_t^k(s,a) \ +$
$\quad \lambda\left(R(s,a) + \gamma \max_{a' \in \mathcal{A}} Q_t^k(s_t(s,a),a')\right).$
7: $\quad\quad$ **if** $(t+1) \mod E = 0$ **then**
8: $\quad\quad\quad Q_{t+1}^k = \frac{1}{K}\sum_{k=1}^K Q_{t+\frac{1}{2}}^k$
9: $\quad\quad$ **else**
10: $\quad\quad\quad Q_{t+1}^k = Q_{t+\frac{1}{2}}^k$
11: $\quad\quad$ **end if**
12: $\quad$ **end for**
13: **end for**
14: **return** $Q_T = \frac{1}{K}\sum_{k=1}^K Q_T^k$

---

### 5.1 Main Convergence Results.

Let $\widetilde{P}_t^k \in \{0,1\}^{|\mathcal{S}||\mathcal{A}| \times |\mathcal{S}|}$ denote the local empirical transition matrix at the $t$-th iteration, defined as

$$\widetilde{P}_t^k(s' \mid s,a) = \mathbf{1}\{s' = s_t^k(s,a)\}.$$

Denoting $\widetilde{P}_i^k V^* \in \mathbb{R}^{|\mathcal{S}||\mathcal{A}| \times 1}$ with the $(s,a)$-th entry as $\widetilde{P}_i^k(s,a)V^* = \sum_{s' \in \mathcal{S}} \widetilde{P}_i^k(s'|s,a)V^*(s')$. Let $\bar{Q}_{t+1} := \frac{1}{K}\sum_{k=1}^K Q_{t+1}^k$. From lines 6, 8, and 10 of Algorithm 1, it follows that

$$\bar{Q}_{t+1} = \frac{1}{K}\sum_{k=1}^K \left((1-\lambda_t)Q_t^k + \lambda_t(R + \gamma\widetilde{P}_t^k V_t^k)\right),$$

where $V_t^k(s) := \max_{a \in \mathcal{A}} Q_t^k(s,a)$ for all $s \in \mathcal{S}$. Define

$$\Delta_{t+1} := Q^* - \bar{Q}_{t+1}, \text{ and } \Delta_0 := Q^* - Q_0. \tag{4}$$

The error iteration $\Delta_t$ is captured in the following lemma.

**Lemma 1** (Error iteration). *For any $t \geq 0$,*

$$\Delta_{t+1} = (1-\lambda)^{t+1}\Delta_0 + \gamma\lambda\sum_{i=0}^t (1-\lambda)^{t-i}\frac{1}{K}\sum_{k=1}^K (\bar{P} - \widetilde{P}_i^k)V^*$$

$$+ \gamma\lambda\sum_{i=0}^t (1-\lambda)^{t-i}\frac{1}{K}\sum_{k=1}^K \widetilde{P}_i^k(V^* - V_i^k). \tag{5}$$

To show the convergence of $\|\Delta_{t+1}\|_\infty$, we bound each of the three terms in the right-hand-side of equation 5. The following lemma is a coarse upper bound of errors.

**Lemma 2.** *Choosing $R(s,a) \in [0,1]$ for each state-action pair $(s,a)$, and choose $0 \le Q_0(s,a) \le \frac{1}{1-\gamma}$ for any $(s,a) \in \mathcal{S} \times \mathcal{A}$, then $0 \le Q_t^k(s,a) \le \frac{1}{1-\gamma}$, $0 \le Q^*(s,a) \le \frac{1}{1-\gamma}$,*

$$\left\| Q^* - Q_t^k \right\|_\infty \le \frac{1}{1-\gamma}, \quad and \quad \left\| V^* - V_t^k \right\|_\infty \le \frac{1}{1-\gamma}, \quad \forall\, t \ge 0, and\ k \in [K]. \tag{6}$$

With the choice of $Q_0$ in Lemma 2, the first term in equation 5 can be bounded as $\left\| (1-\lambda)^{t+1}\Delta_0 \right\|_\infty \le (1-\lambda)^{t+1}\frac{1}{1-\gamma}$. In addition, as detailed in the proof of Lemma 4 and Theorem 1, the boundedness in Lemma 2 enables us to bound the second term in equation 5 via invoking the Hoeffding's inequality. It remains to bound the third term in equation 5, for which we follow the analysis roadmap of Woo et al. (2023) by a two-step procedure that is described in Lemma 3 and Lemma 4. Let

$$\Delta_t^k = Q^* - Q_t^k, \quad and \quad \chi(t) = t - (t \mod E), \tag{7}$$

i.e., $\Delta_t^k$ is the local error of agent $k$, and $\chi(t)$ is the most recent synchronization iteration of $t$.

**Lemma 3.** *If $t \mod E = 0$, then $\max_{s,a} \left\| \frac{1}{K}\sum_{k=1}^K \widetilde{P}_t^k(V^* - V_t) \right\|_\infty \le \left\| \Delta_t \right\|_\infty$. Otherwise,*

$$\left\| \frac{1}{K}\sum_{k=1}^K \widetilde{P}_t^k(V^* - V_t^k) \right\|_\infty \le \left\| \Delta_{\chi(t)} \right\|_\infty + 2\lambda\frac{1}{K}\sum_{k=1}^K \sum_{t'=\chi(t)}^{t-1} \left\| \Delta_{t'}^k \right\|_\infty$$

$$+ \gamma\lambda\frac{1}{K}\sum_{k=1}^K \max_{s,a} \left| \sum_{t'=\chi(t)}^{t-1} \left( \widetilde{P}_{t'}^k(s,a) - \bar{P}(s,a) \right) V^* \right|.$$

*where we use the convention that $\sum_{t'=\chi(t)}^{\chi(t)-1} \left\| \Delta_{t'}^k \right\|_\infty = 0$.*

**Lemma 4.** *Choose $\lambda \le \frac{1}{E}$. For any $\delta \in (0,1)$, with probability at least $(1-\delta)$,*

$$\left\| \Delta_i^k \right\|_\infty \le \left\| \Delta_{\chi(i)} \right\|_\infty + \frac{3\gamma}{1-\gamma}\lambda(E-1)\kappa + \frac{3\gamma}{1-\gamma}\sqrt{\lambda \log\frac{|\mathcal{S}||\mathcal{A}|KT}{\delta}}, \forall\, i \le T, k \in [K]. \tag{8}$$

Both Lemma 3 and Lemma 4 are non-trivial adaptations of the characterization in the analysis of Woo et al. (2023) due to lack of common optimal action for any given state when environments are heterogeneous.

To bound the $\ell_\infty$ norm of the third term in equation 5, we first invoke Lemma 3, followed by Lemma 4. It is worth noting that directly applying Lemma 4 can also lead to a valid error bound yet the resulting bound will not decay as $T$ increases with proper choice of stepsizes.

**Theorem 1** (Convergence). *Choose $E - 1 \le \frac{1-\gamma}{4\gamma\lambda}$ and $\lambda \le \frac{1}{E}$. For any $\delta \in (0,\frac{1}{3})$, with probability at least $1 - 3\delta$, it holds that*

$$\left\| \Delta_T \right\|_\infty \le \frac{4}{(1-\gamma)^2}\exp\left\{ -\frac{1}{2}\sqrt{(1-\gamma)\lambda T} \right\} + \frac{2\gamma^2}{(1-\gamma)^2}(6\lambda^2(E-1)^2 + \lambda(E-1))\kappa$$

$$+ \left( \frac{12\gamma^2\lambda}{(1-\gamma)^2}\sqrt{E-1} + \frac{2\gamma^2\sqrt{\lambda}}{(1-\gamma)^2} \right)\sqrt{\lambda(E-1)\log\frac{|\mathcal{S}||\mathcal{A}|KT}{\delta}}$$

$$+ \frac{2\gamma}{(1-\gamma)^2}\sqrt{\frac{1}{K}\lambda\log\frac{|\mathcal{S}||\mathcal{A}|TK}{\delta}}.$$

The first term of Theorem 1 is the standard error bound in the absence of environmental heterogeneity and sampling noises. The second term arises from environmental heterogeneity. It is clear that when $E = 1$, the environmental heterogeneity does not negatively impact the convergence. The last two terms result from the randomness in sampling.

*Remark* 1 (Eventual zero error). It is common to choose the stepsize $\lambda$ based on the time horizon $T$. Let $\lambda = g(T)$ be a non-increasing function of $T$. As long as $\lambda = g(T)$ decay in $T$, terms 2-4 in Theorem 1 will go to 0 as $T$ increases. In addition, when $\lambda = \omega(1/T)$, the first term will decay to 0.

There is a tradeoff in the convergence rates of the first term and the remaining terms – the slower $\lambda$ decay in $T$ leads to faster decay in the first term but slower in the remaining terms. Forcing these terms to decay around the same speed lead to slow overall convergence. Corollary 1 follows immediately from Theorem 1 via carefully choosing $\lambda$ to balance the decay rates of different terms.

**Corollary 1.** *Choose* $(E-1) \leq \min \frac{1}{\lambda}\{\frac{\gamma}{1-\gamma}, \frac{1}{K}\}$, *and* $\lambda = \frac{4\log^2(TK)}{T(1-\gamma)}$. *Let* $T \geq E$. *For any* $\delta \in (0, \frac{1}{3})$, *with probability at least* $1 - 3\delta$,

$$\|\Delta_T\|_\infty \leq \frac{4}{(1-\gamma)^2 TK} + \frac{36}{(1-\gamma)^3} \frac{\log(TK)}{\sqrt{TK}} \sqrt{\log \frac{|S||A|TK}{\delta}} + \frac{56\log^2(TK)}{(1-\gamma)^3} \frac{E-1}{T} \kappa.$$

*Remark* 2 (Partial linear speedup and the negative impacts of $E > 1$). Intuitively, both terms 1 and 2 decay as if there are $TK$ iterations – a linear speedup. In fact, the decay rate of the sampling noises in Corollary 1, with respect to $TK$, is the minimax optimal up to polylog factors (Vershynin, 2018). The decay of the third term is controlled by environmental heterogeneity when $E > 1$. In sharp contrast to the homogeneous settings, larger $E$ significantly slows down the convergence of this term.

We show in the next subsection that this slow convergence is fundamental.

## 5.2 On the fundamentals of Convergence Slowing Down for $E > 1$.

**Theorem 2.** *Let* $Q_0 = \mathbf{0}$. *For any even* $K \geq 2$, *there exist a collection of* $\{(S, A, P^k, R, \gamma): k \in [K]\}$ *such that, for any synchronization period* $E$ *and time-invariant stepsize* $\lambda \leq \frac{1}{1+\gamma}$,

$$\|\Delta_T\|_\infty = \Omega(E/T),$$

*when* $T/E \in \mathbb{N}$ *and* $T \gtrsim \frac{E}{1-\gamma} \log \frac{1}{1-\gamma}$.

**Proof Sketch.** Below we discuss the key ideas and provide the proof sketch of Theorem 2. The full proof is deferred to Appendix F.

The eventual slow rate convergence is due to the heterogeneous environments $P^k$ regardless of the cardinality of the action space. In particular, we prove the slow rate when the action space is a singleton, in which case the Q-function coincides with the V-function. The process is also known as the Markov reward process. According to Algorithm 1, when $(t+1) \mod E \neq 0$, we have

$$Q_{t+1}^k = ((1-\lambda)I + \lambda\gamma P^k) Q_t^k + \lambda R.$$

Following Algorithm 1, we obtain the following recursion between two synchronization rounds:

$$\Delta_{(r+1)E} = \bar{A}^{(E)}\Delta_{rE} + \left(\left(I - \bar{A}^{(E)}\right) - \left(I + \bar{A}^{(1)} + \dots \bar{A}^{(E-1)}\right)\left(I - \bar{A}^{(1)}\right)\right)Q^*, \qquad (9)$$

where $\bar{A}^{(\ell)} \triangleq \frac{1}{K}\sum_{k=1}^{K}(A^k)^\ell$ and $A^k \triangleq (1-\lambda)I + \lambda\gamma P^k$. While the first term on the right-hand side of equation 9 decays rapidly to zero, the second term is non-vanishing due to environment heterogeneity for $E \geq 2$. Specifically, to ensure the rapid decay of the first term, it is necessary to select a stepsize $\lambda = \widetilde{\Omega}(\frac{1}{rE})$. However, this choice results in the dominating residual error from the second term, which increases linearly with $\lambda E = \widetilde{\Omega}(\frac{1}{r})$.

Next, we instantiate the analyses by constructing the set $P^k$ over a pair of states and an even number of clients with

$$P^{2k-1} = \begin{bmatrix} 1 & 0 \\ 0 & 1 \end{bmatrix}, \quad P^{2k} = \begin{bmatrix} 0 & 1 \\ 1 & 0 \end{bmatrix}, \quad \text{for } k \in \mathbb{N}. \qquad (10)$$

Applying the formula of $\bar{A}^{(\ell)}$ yields the following eigen-decomposition:

$$\bar{A}^{(\ell)} = \alpha_\ell(I - \bar{P}) + \beta_\ell \bar{P},$$

where $\bar{P} = \frac{1}{2}\mathbf{1}\mathbf{1}^\top$, $\alpha_\ell \triangleq \frac{1}{2}(\nu_1^\ell + \nu_2^\ell)$, $\beta_\ell \triangleq \nu_2^\ell$, $\nu_1 \triangleq 1 - (1 + \gamma)\lambda$, and $\nu_2 \triangleq 1 - (1 - \gamma)\lambda$. For this instance of $\mathcal{P}_k$, the error evolution equation 9 reduces to $\Delta_{(r+1)E} = \left(\alpha_E(I - \bar{P}) + \beta_E\bar{P}\right)\Delta_{rE} + \kappa_E(I - \bar{P})Q^*$ with $\kappa_E \triangleq -\frac{\gamma}{2}\left(\frac{1-\nu_2^E}{1-\gamma} - \frac{1-\nu_1^E}{1+\gamma}\right)$, which further yields the following full error recursion:

$$\Delta_{rE} = \left(\alpha_E^r(I - \bar{P}) + \beta_E^r\bar{P}\right)\Delta_0 + \frac{1 - \alpha_E^r}{1 - \alpha_E}\kappa_E(I - \bar{P})Q^*.$$

Starting from $Q_0 = 0$, the error can be decomposed into

$$\Delta_{rE} = \beta_E^r\bar{P}Q^* + \left(\alpha_E^r + \frac{1 - \alpha_E^r}{1 - \alpha_E}\kappa_E\right)(I - \bar{P})Q^*. \tag{11}$$

The two terms of the error are orthogonal and both non-vanishing. Therefore, it remains to lower bound the maximum magnitude of two coefficients irrespective of the stepsize $\lambda$. To this end, we analyze two regimes of $\lambda$ separated by a threshold $\lambda_0 \triangleq \frac{\log r}{(1-\gamma)rE}$:

- Slow rate due to small stepsize when $\lambda \leq \lambda_0$. Since $\beta_E^r$ decreases as $\lambda$ increases,

$$\beta_E^r \geq (1 - (1 - \gamma)\lambda_0)^{rE} = \left(1 - \frac{\log r}{rE}\right)^{rE} \gtrsim \frac{1}{r}.$$

- Slow rate due to environment heterogeneity when $\lambda \geq \lambda_0$. We show that

$$\left|\frac{\kappa_E}{1 - \alpha_E}\right| \geq \gamma^2\frac{\lambda(E - 1)}{4} \gtrsim \gamma^2\frac{\log r}{(1 - \gamma)r}, \qquad \left(1 + \left|\frac{\kappa_E}{1 - \alpha_E}\right|\right)\alpha_E^r \leq \frac{1}{(1 - \gamma^2)r}.$$

We conclude that at least one component of the error in equation 11 must be slower than the rate $\Omega(1/r)$.

*Remark* 3. The explicit calculations are based on a set $\mathcal{P}^k$ over a pair of states. Nevertheless, the evolution equation 9 is generally applicable. Similar analyses can be extended to scenarios involving more than two states, provided that the sequence of matrices $\bar{A}^{(\ell)}$ is simultaneously diagonalizable. For instance, the construction of the transition kernels in equation 10 can be readily extended to multiple states if the set $\mathcal{S}$ can be partitioned into two different classes. The key insight is the non-vanishing residual on the right-hand side of equation 9 when $E \geq 2$ due to the environment heterogeneity.

## 6 Experiments

**Description of the setup.** In our experiments, we consider $K = 5$ agents (Jin et al., 2022), each interacting with an independently and randomly generated $5 \times 5$ maze environment $\langle\mathcal{S}, \mathcal{A}, \mathcal{P}^k, R, \gamma\rangle$ for $k \in \{1, 2, \cdots, 5\}$. The state set $\mathcal{S}$ contains 25 cells that the agent is currently in. The action set contains 4 actions $\mathcal{A} = \{\text{left, up, right, down}\}$. Thus, $|\mathcal{S}| \times |\mathcal{A}| = 100$. We choose $\gamma = 0.99$. For ease of verifying our theory, each entry of the reward $R \in \mathbb{R}^{100}$ is sampled from $\text{Bern}(p = 0.05)$, which slightly departs from a typical maze environment wherein only two state-action pairs have nonzero rewards. We choose this reward so that $\|\Delta_0\|_\infty \approx 100 = \frac{1}{1-\gamma}$, which is the coarse upper bound of $\|\Delta_t\|_\infty$ for all $t$. For each agent $k$, its state transition probability vectors $\mathcal{P}^k$ are constructed on top of standard state transition probability vectors of maze environments incorporated with a drifting probability 0.1 in each non-intentional action as in `WindyCliff` (Jin et al., 2022; Paul et al., 2019). In this way, the environment heterogeneity lies not only in the differences of the non-zero probability values (Jin et al., 2022; Paul et al., 2019) but also in the probability supports (i.e., the locations of non-zero entries). Our construction is more challenging: The environment heterogeneity $\kappa$ as per (2) of our environment construction was calculated to be 1.2. Yet, the largest environment heterogeneity of the `WindyCliff` construction in Jin et al. (2022) is about 0.31.

We choose $Q_0 = \mathbf{0} \in \mathbb{R}^{100}$. All numerical results are based on 5 independent runs to capture the variability. The dark lines represent the mean of the runs, while the shaded areas around each line illustrate the range obtained by adding and subtracting one standard deviation from the mean. The maximum time duration is $T = 20,000$ in our experiment since it is sufficient to capture the characteristics of the training process.

**Two-phase phenomenon.** We plot the evolutions of $\|\Delta_t\|_\infty$ for synchronous federated Q-learning under heterogeneous and homogeneous environments, respectively. Our results show that the sharp two-phase phenomenon mainly arises from environmental heterogeneity rather than sampling noise.

From Figure 2a, it is clear that under the heterogeneous setting, for a given set of constant stepsizes $\lambda \in \{0.9, 0.5, 0.2, 0.1, 0.05\}$, the $\ell_\infty$-norm of $\Delta_t = \bar{Q}_t - Q^*$ decreases to a minimum point and then bounces back rapidly before stabilizing around some fixed errors. Moreover, we can see that different stepsizes give different minimum error. The smaller the stepsize, the smaller the minimum error; however, it takes longer to reach such minimum errors. In sharp contrast, as shown in Figure 2b, there is no drastic bounce when the environments are homogeneous.

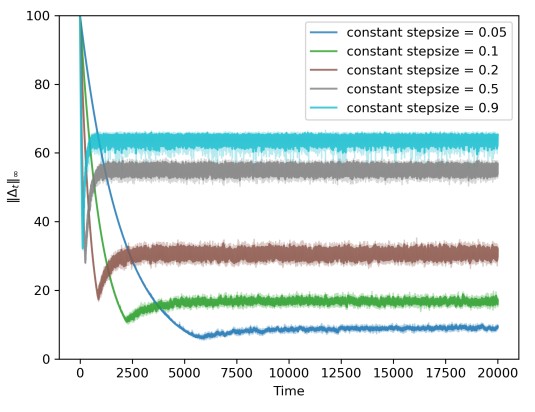 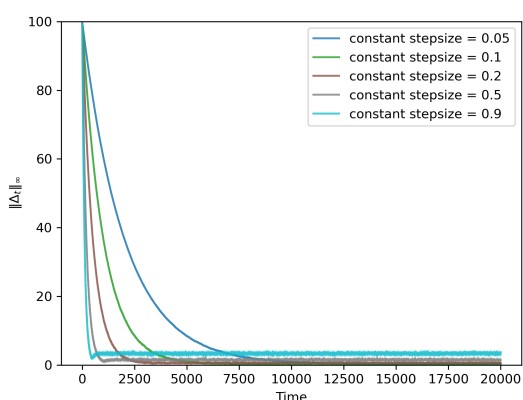

(a) Heterogeneous environments $E = 10$.        (b) Homogeneous environments $E = 10$.

Figure 2: The $\ell_\infty$ error of different constant stepsizes under the heterogeneous and homogenous settings.

A useful practice implication of our results is that: While constant stepsizes are often used in reinforcement learning problems because of the great performance in applications as described in Sutton & Barto (2018), they suffer significant performance degradation in the presence of environmental heterogeneity.

**Impacts of the synchronization period $E$.** Furthermore, we test the impacts of the synchronization period $E$. As shown in Figure 3 and Figure 2a, with $\lambda \in \{0.9, 0.5, 0.2, 0.1, 0.05\}$, as $E$ increases, the final error increases and saturates around 62 in the presence of environmental heterogeneity. For a homogeneous setting (results deferred to Appendix G.1), $E$ does not have a significant impact, which aligns with the observations in the existing literature on the homogeneous settings (Woo et al., 2023; Khodadadian et al., 2022).

**Potential utilization of the two-phase phenomenon.**

As shown in Figures 2a and 3, in the presence of environmental heterogeneity, the smaller the stepsizes, the smaller error $\|\Delta_t\|_\infty$ can reach and less significant of the error bouncing in the second phase.

In our preliminary experiments, we tested small stepsizes $\lambda = 1/T^\alpha$ for $\alpha \in \{0.4, 0.5, \cdots, 1\}$, which eventually lead to small errors yet at the cost of being extremely slow. Among these choices, $\lambda = 1/\sqrt{T}$ has the fastest convergence performance yet is still $\approx 24$ up to iteration 20,000.

Let $t_0$ be the iteration at which the error trajectory $\|\Delta_t\|_\infty$ switches from phase 1 to phase 2. Provided that $t_0$ can be estimated, choosing different stepsizes for the two phases can lead to faster overall convergence, compared with using the same stepsize throughout.

Figure 4 illustrates two-phase training with different phase 1 stepsizes and phase 2 stepsize $\lambda = 1/\sqrt{T}$ compared with using $\lambda = 1/\sqrt{T}$ throughout. Overall, using $\lambda = 1/\sqrt{T}$ throughout leads to the slowest convergence, highlighting the benefits of the two-phase training strategy. Among all two-phase stepsize choices, the stepsize of 0.05 in the first phase results in a longer phase 1 duration ($t_0 = 5550$) but the lowest

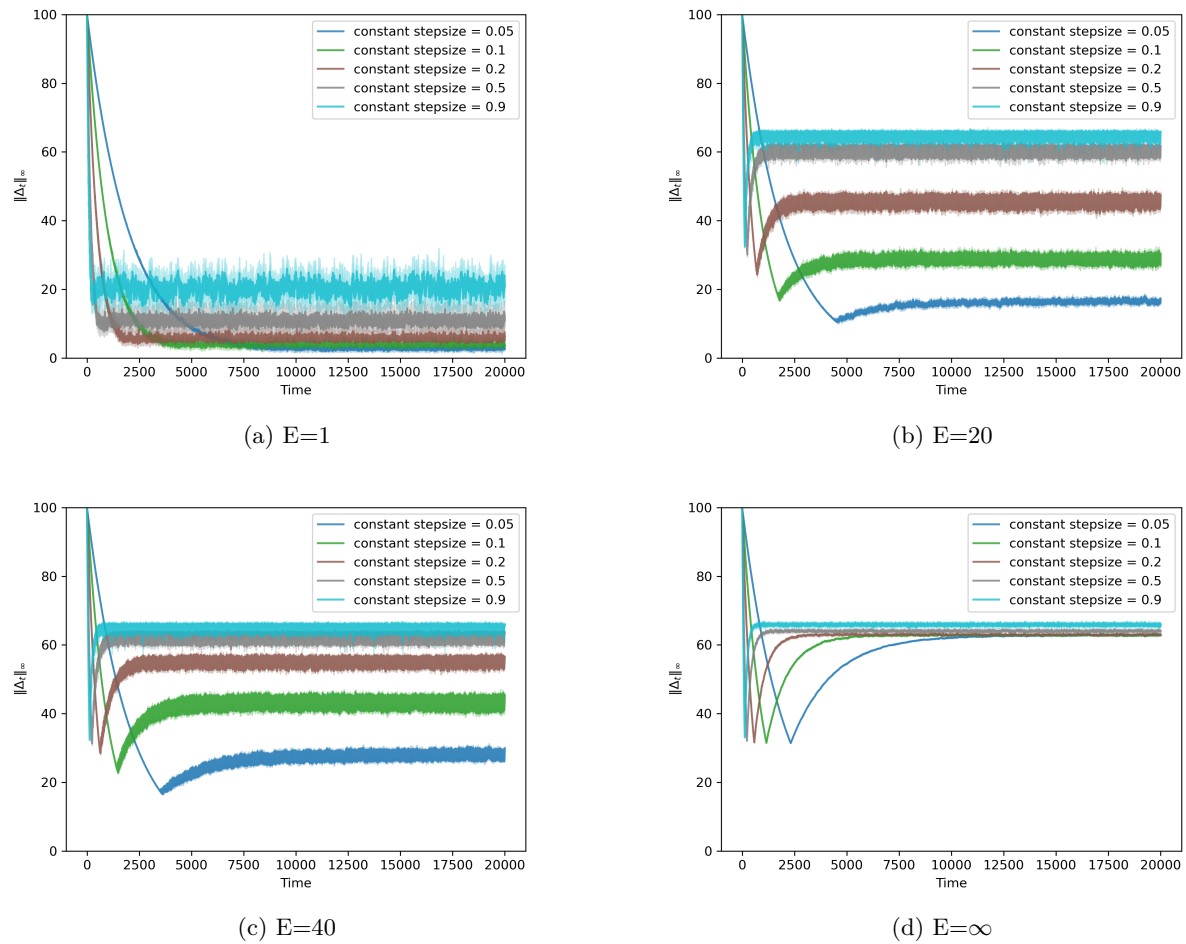

Figure 3: Heterogeneous environments with varying $E$.

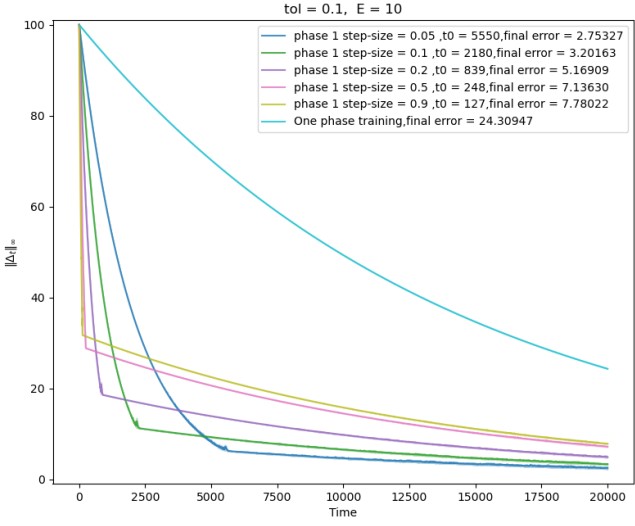

Figure 4: Choosing different stepsizes for phases 1 and 2 leads to faster overall convergence. $E = 10$.

final error (2.75327), suggesting a better convergence. We further test the convergence performance with respect to different target error levels, details can be found in Appendix G.2.

We leave the estimation and characterization of $t_0$ for future work.

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

# Appendices

## A  Proof of Lemma 1

The update of $\Delta_{t+1}$ is as follows:

$$
\begin{aligned}
\Delta_{t+1} &= Q^* - \bar{Q}_{t+1} \\
&= \frac{1}{K} \sum_{k=1}^{K} (Q^* - ((1-\lambda)Q_t^k + \lambda(R + \gamma \widetilde{P}_t^k Q_t^k))) \\
&= \frac{1}{K} \sum_{k=1}^{K} ((1-\lambda)(Q^* - Q_t^k) + \lambda(Q^* - R - \gamma \widetilde{P}_t^k V_t^k)) \\
&= (1-\lambda)\Delta_t + \gamma\lambda \frac{1}{K} \sum_{k=1}^{K} (\bar{P}V^* - \widetilde{P}_t^k V_t^k) \\
&= (1-\lambda)\Delta_t + \frac{\gamma\lambda}{K} \sum_{k=1}^{K} (\bar{P} - \widetilde{P}_t^k)V^* + \frac{\gamma\lambda}{K} \sum_{k=1}^{K} \widetilde{P}_t^k (V^* - V_t^k) \\
&= (1-\lambda)^{t+1}\Delta_0 + \gamma\lambda \sum_{i=0}^{t} (1-\lambda)^{t-i} \frac{1}{K} \sum_{k=1}^{K} (\bar{P} - \widetilde{P}_i^k)V^* \\
&\quad + \gamma\lambda \sum_{i=0}^{t} (1-\lambda)^{t-i} \frac{1}{K} \sum_{k=1}^{K} \widetilde{P}_i^k (V^* - V_i^k),
\end{aligned}
$$

recalling that $\Delta_0 = Q^* - Q_0$.

## B  Proof of Lemma 2

We first show $0 \le Q_t^k(s,a) \le \frac{1}{1-\gamma}$ by inducting on $t$. When $t = 0$, this is true by the choice of $Q_0$. Suppose that $0 \le Q_{t-1}^k(s,a) \le \frac{1}{1-\gamma}$ for any state-action pair $(s,a)$ and any client $k$. Let's focus on time $t$. When $t$ is not a synchronization iteration (i.e., $t+1 \mod \ne 0$), we have

$$
\begin{aligned}
Q_t^k(s,a) &= (1-\lambda)Q_{t-1}^k(s,a) + \lambda(R(s,a) + \gamma \widetilde{P}_t^k V_{t-1}^k(s)) \\
&\le \frac{1-\lambda}{1-\gamma} + \lambda(R(s,a) + \gamma \widetilde{P}_t^k V_{t-1}^k(s)) \\
&\overset{(a)}{\le} \frac{1-\lambda}{1-\gamma} + \lambda(1 + \frac{\gamma}{1-\gamma}) \\
&\le \frac{1}{1-\gamma} - \frac{\lambda}{1-\gamma} + \frac{\lambda}{1-\gamma} \\
&= \frac{1}{1-\gamma},
\end{aligned}
$$

where inequality (a) holds because for any $s$, $V_{t-1}^k(s) = \max_{a \in \mathcal{A}} Q_{t-1}^k(s,a) \le \frac{1}{1-\gamma}$ by the inductive hypothesis, and $\widetilde{P}_t^k\|_1 = 1$. Similarly, we can show the case when $t$ is a synchronization iteration.

With the above argument, we can also show that $0 \le Q^*(s,a) \le \frac{1}{1-\gamma}$ for any state-action pair $(s,a)$. Therefore, we have that $\left\| Q^* - Q_t^k \right\|_\infty \le \frac{1}{1-\gamma}$.

Next, we show that bound on $\left\|V^* - V_t^k\right\|_\infty$.

$$\left\|V^* - V_t^k\right\|_\infty = \max_{s \in \mathcal{S}} \left|V^*(s) - V_t^k(s)\right|$$

$$= \max_{s \in \mathcal{S}} \left|\max_{a \in \mathcal{A}} Q^*(s, a) - \max_{a' \in \mathcal{A}} Q_t^k(s, a')\right|$$

$$\leq \max_{s \in \mathcal{S}, a \in \mathcal{A}} \left|Q^*(s, a) - Q_t^k(s, a)\right|$$

$$= \left\|Q^* - Q_t^k\right\|_\infty$$

$$\leq \frac{1}{1 - \gamma}.$$

## C  Proof of Lemma 3

When $t \mod E = 0$, i.e., $i$ is a synchronization round, $Q_t^k = Q_t^{k'}$ for any pair of agents $k, k' \in [K]$. Hence,

$$\frac{1}{K} \sum_{k=1}^{K} \widetilde{P}_t^k(s, a)(V^* - V_t) = \left(\frac{1}{K} \sum_{k=1}^{K} \widetilde{P}_t^k(s, a)\right)(V^* - V_t)$$

$$\leq \|\frac{1}{K} \sum_{k=1}^{K} \widetilde{P}_t^k(s, a)\|_1 \|V^* - V_t\|_\infty$$

$$\leq \|Q^* - Q_t\|_\infty$$

$$= \|\Delta_t\|_\infty. \tag{12}$$

For general $t$, we have

$$\left\|\frac{1}{K} \sum_{k=1}^{K} \widetilde{P}_t^k(V^* - V_t^k)\right\|_\infty = \left\|\frac{1}{K} \sum_{k=1}^{K} \widetilde{P}_t^k(V^* - V_{\chi(t)}^k + V_{\chi(t)}^k - V_i^k)\right\|_\infty$$

$$\leq \left\|\frac{1}{K} \sum_{k=1}^{K} \widetilde{P}_t^k(V^* - V_{\chi(t)}^k)\right\|_\infty + \left\|\frac{1}{K} \sum_{k=1}^{K} \widetilde{P}_t^k(V_{\chi(t)}^k - V_t^k)\right\|_\infty$$

$$\leq \left\|\Delta_{\chi(t)}\right\|_\infty + \left\|\frac{1}{K} \sum_{k=1}^{K} \widetilde{P}_t^k(V_{\chi(t)}^k - V_t^k)\right\|_\infty \quad \text{by equation 12}$$

$$\leq \left\|\Delta_{\chi(t)}\right\|_\infty + \frac{1}{K} \sum_{k=1}^{K} \left\|V_{\chi(t)}^k - V_t^k\right\|_\infty. \tag{13}$$

For any state $s \in \mathcal{S}$, we have

$$V_t^k(s) - V_{\chi(t)}^k(s)$$

$$= Q_t^k(s, a_t^k(s)) - Q_{\chi(t)}^k(s, a_{\chi(t)}^k(s))$$

$$\overset{(a)}{\leq} Q_t^k(s, a_t^k(s)) - Q_{\chi(t)}^k(s, a_t^k(s))$$

$$= Q_t^k(s, a_t^k(s)) - Q_{t-1}^k(s, a_t^k(s)) + Q_{t-1}^k(s, a_t^k(s)) - Q_{t-2}^k(s, a_t^k(s))$$

$$+ \cdots + Q_{\chi(t)+1}^k(s, a_t^k(s)) - Q_{\chi(t)}^k(s, a_t^k(s)). \tag{14}$$

where inequality (a) holds because $Q_{\chi(t)}^k(s, a_t^k(s)) \leq Q_{\chi(t)}^k(s, a_{\chi(t)}^k(s))$.

For each $t'$ such that $\chi(t) \le t' \le t$, it holds that,

$$
\begin{aligned}
&Q_{t'+1}^k(s, a_t^k(s)) - Q_{t'}^k(s, a_t^k(s)) \\
&= (1-\lambda)Q_{t'}^k(s, a_t^k(s)) + \lambda(R(s, a_t^k(s)) + \gamma \widetilde{P}_{t'}^k(s, a_t^k(s))V_{t'}^k) - Q_{t'}^k(s, a_t^k(s)) \\
&\overset{(a)}{=} -\lambda Q_{t'}^k(s, a_t^k(s)) + \lambda \left( Q^*(s, a_t^k(s)) - R(s, a_t^k(s)) - \gamma \bar{P}(s, a_t^k(s))V^* + R(s, a_t^k(s)) + \gamma \widetilde{P}_{t'}^k(s, a_t^k(s))V_{t'}^k \right) \\
&= \lambda \Delta_{t'}^k(s, a_t^k(s)) + \gamma \lambda \left( (\widetilde{P}_{t'}^k(s, a_t^k(s)) - \bar{P}(s, a_t^k(s)))V^* + \widetilde{P}_{t'}^k(s, a_t^k(s))(V_{t'}^k - V^*) \right) \\
&\le 2\lambda \left\| \Delta_{t'}^k \right\|_\infty + \gamma \lambda \left( \widetilde{P}_{t'}^k(s, a_t^k(s)) - \bar{P}(s, a_t^k(s)) \right) V^*,
\end{aligned}
$$

where equality (a) follows from the Bellman equation equation 3. Thus,

$$
\begin{aligned}
V_t^k(s) - V_{\chi(t)}^k(s) &\le \sum_{t'=\chi(t)}^{t-1} Q_{t'+1}^k(s, a_t^k(s)) - Q_{t'}^k(s, a_t^k(s)) \\
&= 2\lambda \sum_{t'=\chi(t)}^{t-1} \left\| \Delta_{t'}^k \right\|_\infty + \gamma\lambda \sum_{t'=\chi(t)}^{t-1} \left( \widetilde{P}_{t'}^k(s, a_t^k(s)) - \bar{P}(s, a_t^k(s)) \right) V^*.
\end{aligned}
\tag{15}
$$

Similarly, we have

$$
\begin{aligned}
V_t^k(s) - V_{\chi(t)}^k(s) &\ge \sum_{t'=\chi(t)}^{t-1} Q_{t'+1}^k(s, a_{\chi(t)}^k(s)) - Q_{t'}^k(s, a_{\chi(t)}^k(s)) \\
&\ge -2\lambda \sum_{t'=\chi(t)}^{t-1} \left\| \Delta_{t'}^k \right\|_\infty + \gamma\lambda \sum_{t'=\chi(t)}^{t-1} \left( \widetilde{P}_{t'}^k(s, a_{\chi(t)}^k(s)) - \bar{P}(s, a_{\chi(t)}^k(s)) \right) V^*.
\end{aligned}
\tag{16}
$$

Plugging the bounds in equation 15 and in equation 16 back into equation 13, we get

$$
\begin{aligned}
\left\| \frac{1}{K} \sum_{k=1}^K \widetilde{P}_t^k(V^* - V_t^k) \right\|_\infty &\le \left\| \Delta_{\chi(t)} \right\|_\infty + \frac{1}{K} \sum_{k=1}^K \left\| V_{\chi(t)}^k - V_t^k \right\|_\infty \\
&\le \left\| \Delta_{\chi(t)} \right\|_\infty + 2\lambda \frac{1}{K} \sum_{k=1}^K \sum_{t'=\chi(t)}^{t-1} \left\| \Delta_{t'}^k \right\|_\infty \\
&\quad + \gamma\lambda \frac{1}{K} \sum_{k=1}^K \max_{s,a} \left\| \sum_{t'=\chi(t)}^{t-1} \left( \widetilde{P}_{t'}^k(s, a) - \bar{P}(s, a) \right) V^* \right\|_\infty .
\end{aligned}
$$

## D  Proof of Lemma 4

When $i \mod E = 0$, then $\Delta_i^k = \Delta_{\chi(i)}$. When $i \mod E \ne 0$, we have

$$
\begin{aligned}
Q_i^k &= (1-\lambda)Q_{i-1}^k + \lambda \left( R + \gamma \widetilde{P}_{i-1}^k V_{i-1}^k \right) \\
&= (1-\lambda)Q_{i-1}^k + \lambda \left( Q^* - R - \gamma \bar{P}V^* + R + \gamma \widetilde{P}_{i-1}^k V_{i-1}^k \right).
\end{aligned}
$$

So,

$$
\begin{aligned}
\Delta_i^k &= (1-\lambda)\Delta_{i-1}^k + \lambda\gamma\left(\bar{P}V^* - \widetilde{P}_{i-1}^k V_{i-1}^k\right)\\
&= (1-\lambda)\Delta_{i-1}^k + \lambda\gamma(\bar{P} - \widetilde{P}_{i-1}^k)V^* + \lambda\gamma\widetilde{P}_{i-1}^k(V^* - V_{i-1}^k)\\
&\leq (1-\lambda)^{i-\chi(i)}\Delta_{\chi(i)} + \gamma\lambda\sum_{j=\chi(i)}^{i-1}(1-\lambda)^{i-j-1}(\bar{P} - \widetilde{P}_j^k)V^*\\
&\quad + \gamma\lambda\sum_{j=\chi(i)}^{i-1}(1-\lambda)^{i-j-1}\widetilde{P}_j^k(V^* - V_j^k).
\end{aligned}
\tag{17}
$$

For any state-action pair $(s,a)$,

$$
|(1-\lambda)^{i-\chi(i)}\Delta_{\chi(i)}(s,a)| \leq (1-\lambda)^{i-\chi(i)}\left\|\Delta_{\chi(i)}\right\|_\infty.
\tag{18}
$$

By invoking Hoeffding's inequality, for any given $\delta \in \delta \in (0,1)$, with probability at least $1-\delta$, it holds that

$$
\begin{aligned}
\left|\gamma\lambda\sum_{j=\chi(i)}^{i-1}(1-\lambda)^{i-j-1}(\bar{P} - \widetilde{P}_j^k)V^*\right| &\leq \frac{\gamma}{1-\gamma}\lambda\sum_{j=\chi(i)}^{i-1}(1-\lambda)^{i-1-j}\kappa + \frac{\gamma}{1-\gamma}\sqrt{\lambda\log\frac{|\mathcal{S}||\mathcal{A}|KT}{\delta}}\\
&\leq \frac{\gamma}{1-\gamma}\lambda(E-1)\kappa + \frac{\gamma}{1-\gamma}\sqrt{\lambda\log\frac{|\mathcal{S}||\mathcal{A}|KT}{\delta}},
\end{aligned}
\tag{19}
$$

for all $(s,a) \in \mathcal{S} \times \mathcal{A}, i \in [T], k \in [K]$. In addition, we have

$$
\left|\gamma\lambda\sum_{j=\chi(i)}^{i-1}(1-\lambda)^{i-j-1}\widetilde{P}_j^k(V^* - V_j^k)\right| \leq \gamma\lambda\sum_{j=\chi(i)}^{i-1}(1-\lambda)^{i-j-1}\left\|\Delta_j^k\right\|_\infty.
\tag{20}
$$

Combining the bounds in equation 18, equation 19, and equation 20, we get

$$
\begin{aligned}
\left\|\Delta_i^k\right\|_\infty &\leq (1-\lambda)^{i-\chi(i)}\left\|\Delta_{\chi(i)}\right\|_\infty + \frac{\gamma}{1-\gamma}\lambda(E-1)\kappa + \frac{\gamma}{1-\gamma}\sqrt{\lambda\log\frac{|\mathcal{S}||\mathcal{A}|KT}{\delta}}\\
&\quad + \gamma\lambda\sum_{j=\chi(i)}^{i-1}(1-\lambda)^{i-j-1}\left\|\Delta_j^k\right\|_\infty\\
&\leq (1-(1-\gamma)\lambda)^{i-\chi(i)}\left\|\Delta_{\chi(t)}\right\|_\infty\\
&\quad + (1+\gamma\lambda)^{i-\chi(i)}\left(\frac{\gamma}{1-\gamma}\lambda(E-1)\kappa + \frac{\gamma}{1-\gamma}\sqrt{\lambda\log\frac{|\mathcal{S}||\mathcal{A}|KT}{\delta}}\right),
\end{aligned}
\tag{21}
$$

where the last inequality can be shown via inducting on $i - \chi(i) \in \{0, \cdots, E-1\}$. When $\lambda \leq \frac{1}{E}$,

$$
(1+\gamma\lambda)^{i-\chi(i)} \leq (1+\lambda)^E \leq (1+1/E)^E \leq e \leq 3.
$$

We get

$$
\left\|\Delta_i^k\right\|_\infty \leq \left\|\Delta_{\chi(i)}\right\|_\infty + 3\frac{\gamma}{1-\gamma}\lambda(E-1)\kappa + 3\frac{\gamma}{1-\gamma}\sqrt{\lambda\log\frac{|\mathcal{S}||\mathcal{A}|KT}{\delta}}.
$$

## E  Proof of Theorem 1

By Lemma 1,

$$
\Delta_{t+1} = (1-\lambda)^{t+1}\Delta_0 + \sum_{i=0}^{t}(1-\lambda)^i\frac{\gamma\lambda}{K}\sum_{k=1}^{K}(\bar{P} - \widetilde{P}_{t-i}^k)V^* + \sum_{i=0}^{t}(1-\lambda)^i\frac{\gamma\lambda}{K}\sum_{k=1}^{K}\widetilde{P}_{t-i}^k(V^* - V_{t-i}^k).
$$

Taking the $\ell_\infty$ norm on both sides, we get

$$\|\Delta_{t+1}\|_\infty \leq (1-\lambda)^{t+1}\|\Delta_0\|_\infty + \left\|\sum_{i=0}^{t}(1-\lambda)^i\lambda\gamma\frac{1}{K}\sum_{k=1}^{K}(\bar{P}-\widetilde{P}_{t-i}^k)V^*\right\|_\infty$$
$$+ \sum_{i=0}^{t}(1-\lambda)^i\lambda\gamma\left\|\frac{1}{K}\sum_{k=1}^{K}\widetilde{P}_{t-i}^k(V^*-V_{t-i}^k)\right\|_\infty.$$

We bound the three terms in the right-hand-side of the above-displayed equation separately. Since $0 \leq Q_0(s,a) \leq \frac{1}{1-\gamma}$, the first term can be bounded as

$$(1-\lambda)^{t+1}\|\Delta_0\|_\infty \leq (1-\lambda)^{t+1}\frac{1}{1-\gamma}. \tag{22}$$

To bound the second term $\left\|\sum_{i=0}^{t}(1-\lambda)^i\lambda\gamma\frac{1}{K}\sum_{k=1}^{K}(\bar{P}-\widetilde{P}_{t-i}^k)V^*\right\|_\infty$, we have

$$\sum_{i=0}^{t}(1-\lambda)^i\lambda\gamma\frac{1}{K}\sum_{k=1}^{K}(\bar{P}-\widetilde{P}_{t-i}^k)V^* = \sum_{i=0}^{t}(1-\lambda)^i\lambda\gamma\frac{1}{K}\sum_{k=1}^{K}(P^k-\widetilde{P}_{t-i}^k)V^*$$
$$= \frac{1}{K}\sum_{k=1}^{K}\sum_{i=0}^{t}(1-\lambda)^i\lambda\gamma(P^k-\widetilde{P}_{t-i}^k)V^*.$$

Let $X_{i,k}(s,a)) = \frac{1}{K}\gamma\lambda(1-\lambda)^i(P^k-\widetilde{P}_{t-i}^k)V^*$. It is easy to see that $\mathbb{E}[X_{i,k}(s,a)] = 0$ for all $(s,a)$. By Lemma 2, we have $|X_{i,k}| \leq \frac{2}{K(1-\gamma)}\gamma\lambda(1-\lambda)^i$ for all $(s,a)$. Since the sampling across clients and across iterations are independent, via invoking Hoeffding's inequality, for any given $\delta \in (0,1)$, with probability at least $1-\delta$,

$$\left\|\sum_{i=0}^{t}(1-\lambda)^i\lambda\gamma\frac{1}{K}\sum_{k=1}^{K}(\bar{P}-\widetilde{P}_{t-i}^k)V^*\right\|_\infty \leq \frac{\gamma}{1-\gamma}\sqrt{\frac{1}{K}\lambda\log\frac{|\mathcal{S}||\mathcal{A}|TK}{\delta}}. \tag{23}$$

To bound the third term $\sum_{i=0}^{t}(1-\lambda)^i\lambda\gamma\left\|\frac{1}{K}\sum_{k=1}^{K}\widetilde{P}_{t-i}^k(V^*-V_{t-i}^k)\right\|_\infty$, following the roadmap of Woo et al. (2023), we divide the summation into two parts as follows. For any $\beta E \leq t \leq T$, we have

$$\sum_{i=0}^{t}(1-\lambda)^i\lambda\gamma\left\|\frac{1}{K}\sum_{k=1}^{K}\widetilde{P}_{t-i}^k(V^*-V_{t-i}^k)\right\|_\infty$$
$$= \sum_{i=0}^{t}(1-\lambda)^{t-i}\lambda\gamma\left\|\frac{1}{K}\sum_{k=1}^{K}\widetilde{P}_i^k(V^*-V_i^k)\right\|_\infty$$
$$= \sum_{i=0}^{\chi(t)-\beta E}(1-\lambda)^{t-i}\lambda\gamma\left\|\frac{1}{K}\sum_{k=1}^{K}\widetilde{P}_i^k(V^*-V_i^k)\right\|_\infty + \sum_{i=\chi(t)-\beta E+1}^{t}(1-\lambda)^{t-i}\lambda\gamma\left\|\frac{1}{K}\sum_{k=1}^{K}\widetilde{P}_i^k(V^*-V_i^k)\right\|_\infty$$
$$\leq \frac{\gamma}{1-\gamma}(1-\lambda)^{t-\chi(t)+\beta E} + \sum_{i=\chi(t)-\beta E+1}^{t}(1-\lambda)^{t-i}\lambda\gamma\left\|\frac{1}{K}\sum_{k=1}^{K}\widetilde{P}_i^k(V^*-V_i^k)\right\|_\infty.$$

By Lemma 3,

$$\sum_{i=\chi(t)-\beta E+1}^{t} (1-\lambda)^{t-i} \lambda\gamma \left\| \frac{1}{K} \sum_{k=1}^{K} \widetilde{P}_i^k (V^* - V_i^k) \right\|_\infty$$

$$\leq \sum_{i=\chi(t)-\beta E+1}^{t} (1-\lambda)^{t-i} \lambda\gamma \left( \left\| \Delta_{\chi(i)} \right\|_\infty + 2\lambda \frac{1}{K} \sum_{k=1}^{K} \sum_{j=\chi(i)}^{i-1} \left\| \Delta_{t'}^k \right\|_\infty \right.$$

$$\left. + \gamma\lambda \frac{1}{K} \sum_{k=1}^{K} \max_{s,a} \left| \sum_{j=\chi(i)}^{i-1} \left( \widetilde{P}_j^k(s,a) - \bar{P}(s,a) \right) V^* \right| \right).$$

Since $\widetilde{P}_j^k(s,a)$'s are independent across time $j$ and across state action pair $(s,a)$, and $|\widetilde{P}_j^k(s,a) - \bar{P}(s,a)V^*| \leq \frac{1}{1-\gamma}$ (from Lemma 2), with Hoeffding's inequality and union bound, we get for any $\delta \in (0,1)$, with probability at least $1-\delta$,

$$\left| \sum_{j=\chi(i)}^{i-1} \left( \widetilde{P}_j^k(s,a) - \bar{P}(s,a) \right) V^* \right| \leq (E-1)\frac{1}{1-\gamma}\kappa + \frac{1}{1-\gamma}\sqrt{(E-1)\log\frac{|\mathcal{S}||\mathcal{A}|KT}{\delta}} \tag{24}$$

for all $(s,a) \in \mathcal{S} \times \mathcal{A}$, $k \in K$, and $i$. By Lemma 4, with probability at least $(1-\delta)$, we have

$$\sum_{i=\chi(t)-\beta E+1}^{t} (1-\lambda)^{t-i} \lambda\gamma 2\lambda \frac{1}{K} \sum_{k=1}^{K} \sum_{j=\chi(i)}^{i-1} \left\| \Delta_j^k \right\|_\infty$$

$$\leq 2\lambda^2\gamma \sum_{i=\chi(t)-\beta E+1}^{t} (1-\lambda)^{t-i} \frac{1}{K} \sum_{k=1}^{K} \sum_{j=\chi(i)}^{i-1} \left( \left\| \Delta_{\chi(i)} \right\|_\infty + 3\frac{\gamma}{1-\gamma}\lambda(E-1)\kappa + 3\frac{\gamma}{1-\gamma}\sqrt{\lambda\log\frac{|\mathcal{S}||\mathcal{A}|KT}{\delta}} \right)$$

$$\leq 2\lambda\gamma(E-1) \max_{\chi(t)-\beta E\leq i\leq t} \left\| \Delta_{\chi(i)} \right\|_\infty + \frac{6\gamma^2\lambda^2}{1-\gamma}(E-1)^2\kappa + \frac{6\gamma^2\lambda}{1-\gamma}(E-1)\sqrt{\lambda\log\frac{|\mathcal{S}||\mathcal{A}|KT}{\delta}}.$$

Thus, with probability at least $(1-2\delta)$,

$$\sum_{i=\chi(t)-\beta E+1}^{t} (1-\lambda)^{t-i} \lambda\gamma \left\| \frac{1}{K} \sum_{k=1}^{K} \widetilde{P}_i^k (V^* - V_i^k) \right\|_\infty$$

$$\leq \gamma \max_{\chi(t)-\beta E\leq i\leq t} \left\| \Delta_{\chi(i)} \right\|_\infty + 2\lambda\gamma(E-1) \max_{\chi(t)-\beta E\leq i\leq t} \left\| \Delta_{\chi(i)} \right\|_\infty + \frac{6\gamma^2\lambda^2}{1-\gamma}(E-1)^2\kappa$$

$$+ \frac{6\gamma^2\lambda}{1-\gamma}(E-1)\sqrt{\lambda\log\frac{|\mathcal{S}||\mathcal{A}|KT}{\delta}}$$

$$+ \sum_{i=\chi(t)-\beta E+1}^{t} (1-\lambda)^{t-i} \lambda\gamma \left( \frac{\gamma\lambda}{1-\gamma}(E-1)\kappa + \frac{\gamma\lambda}{1-\gamma}\sqrt{(E-1)\log\frac{|\mathcal{S}||\mathcal{A}|KT}{\delta}} \right)$$

$$= \gamma(1+2\lambda(E-1)) \max_{\chi(t)-\beta E\leq i\leq t} \left\| \Delta_{\chi(i)} \right\|_\infty + \frac{\gamma^2}{1-\gamma}(6\lambda^2(E-1)^2 + \lambda(E-1))\kappa$$

$$+ \frac{\gamma^2\lambda}{1-\gamma}\sqrt{(E-1)\log\frac{|\mathcal{S}||\mathcal{A}|KT}{\delta}} + \frac{6\gamma^2\lambda}{1-\gamma}(E-1)\sqrt{\lambda\log\frac{|\mathcal{S}||\mathcal{A}|KT}{\delta}}.$$

The third term can be bounded as

$$\sum_{i=0}^{t}(1-\lambda)^i \lambda\gamma \left\| \frac{1}{K}\sum_{k=1}^{K}\widetilde{P}_i^k(V^* - V_i^k)\right\|_\infty$$

$$\leq \frac{\gamma}{1-\gamma}(1-\lambda)^{t-\chi(t)+\beta E} + \gamma(1+2\lambda(E-1))\max_{\chi(t)-\beta E\leq i\leq t}\left\|\Delta_{\chi(i)}\right\|_\infty + \frac{\gamma^2}{1-\gamma}(6\lambda^2(E-1)^2 + \lambda(E-1))\kappa$$

$$+\frac{\gamma^2\lambda}{1-\gamma}\sqrt{(E-1)\log\frac{|\mathcal{S}||\mathcal{A}|KT}{\delta}} + \frac{6\gamma^2\lambda}{1-\gamma}(E-1)\sqrt{\lambda\log\frac{|\mathcal{S}||\mathcal{A}|KT}{\delta}}. \tag{25}$$

Combing the bounds for terms 1, 2, and 3, we get the following recursion holds for all rounds $T$ with probability at least $(1-3\delta)$:

$$\left\|\Delta_{t+1}\right\|_\infty \leq (1-\lambda)^{t+1}\frac{1}{1-\gamma} + \frac{\gamma}{1-\gamma}\sqrt{\frac{1}{K}\lambda\log\frac{|S||A|TK}{\delta}} + \frac{\gamma}{1-\gamma}(1-\lambda)^{t-\chi(t)+\beta E}$$

$$+\gamma(1+2\lambda(E-1))\max_{\chi(t)-\beta E\leq i\leq t}\left\|\Delta_{\chi(i)}\right\|_\infty + \frac{\gamma^2}{1-\gamma}(6\lambda^2(E-1)^2 + \lambda(E-1))\kappa$$

$$+\frac{\gamma^2\lambda}{1-\gamma}\sqrt{(E-1)\log\frac{|\mathcal{S}||\mathcal{A}|KT}{\delta}} + \frac{6\gamma^2\lambda}{1-\gamma}(E-1)\sqrt{\lambda\log\frac{|\mathcal{S}||\mathcal{A}|KT}{\delta}}$$

$$\leq \gamma(1+2\lambda(E-1))\max_{\chi(t)-\beta E\leq i\leq t}\left\|\Delta_{\chi(i)}\right\|_\infty + \frac{2}{1-\gamma}(1-\lambda)^{\beta E} + \frac{\gamma^2}{1-\gamma}(6\lambda^2(E-1)^2 + \lambda(E-1))\kappa$$

$$+\frac{\gamma^2\lambda}{1-\gamma}\sqrt{(E-1)\log\frac{|\mathcal{S}||\mathcal{A}|KT}{\delta}} + \frac{6\gamma^2\lambda}{1-\gamma}(E-1)\sqrt{\lambda\log\frac{|\mathcal{S}||\mathcal{A}|KT}{\delta}}$$

$$+\frac{\gamma}{1-\gamma}\sqrt{\frac{1}{K}\lambda\log\frac{|S||A|TK}{\delta}}.$$

Let

$$\rho := \frac{2}{1-\gamma}(1-\lambda)^{\beta E} + \frac{\gamma^2}{1-\gamma}(6\lambda^2(E-1)^2 + \lambda(E-1))\kappa$$

$$+\frac{\gamma^2\lambda}{1-\gamma}\sqrt{(E-1)\log\frac{|\mathcal{S}||\mathcal{A}|KT}{\delta}} + \frac{6\gamma^2\lambda}{1-\gamma}(E-1)\sqrt{\lambda\log\frac{|\mathcal{S}||\mathcal{A}|KT}{\delta}}$$

$$+\frac{\gamma}{1-\gamma}\sqrt{\frac{1}{K}\lambda\log\frac{|S||A|TK}{\delta}}. \tag{26}$$

With the assumption that $\lambda \leq \frac{1-\gamma}{4\gamma(E-1)}$, the above recursion can be written as

$$\left\|\Delta_{t+1}\right\|_\infty \leq \frac{1+\gamma}{2}\max_{\chi(t)-\beta E\leq i\leq t}\left\|\Delta_{\chi(i)}\right\|_\infty + \rho.$$

Unrolling the above recursion $L$ times where $L\beta E \leq t \leq T$, we obtain that

$$\left\|\Delta_{t+1}\right\|_\infty \leq (\frac{1+\gamma}{2})^L\max_{\chi(t)-L\beta E\leq i\leq t}\left\|\Delta_{\chi(i)}\right\|_\infty + \sum_{i=0}^{L-1}(\frac{1+\gamma}{2})^i\rho$$

$$\leq (\frac{1+\gamma}{2})^L\frac{1}{1-\gamma} + \frac{2}{1-\gamma}\rho.$$

Choosing $\beta = \frac{1}{E}\sqrt{\frac{(1-\gamma)T}{2\lambda}}$, $L = \sqrt{\frac{\lambda T}{1-\gamma}}$, $t + 1 = T$, we get

$$
\begin{aligned}
\|\Delta_T\|_\infty \leq\ & \frac{1}{1-\gamma}(\frac{1+\gamma}{2})\sqrt{\frac{\lambda T}{1-\gamma}} + \frac{2}{1-\gamma}\left(\frac{2}{1-\gamma}(1-\lambda)^{\beta E} + \frac{\gamma^2}{1-\gamma}(6\lambda^2(E-1)^2 + \lambda(E-1))\kappa\right. \\
& + \left.\left(\frac{6\gamma^2\lambda}{1-\gamma}\sqrt{E-1} + \frac{\gamma^2\sqrt{\lambda}}{1-\gamma}\right)\sqrt{\lambda(E-1)\log\frac{|\mathcal{S}||\mathcal{A}|KT}{\delta}} + \frac{\gamma}{1-\gamma}\sqrt{\frac{1}{K}\lambda\log\frac{|S||A|TK}{\delta}}\right) \\
\leq\ & \frac{1}{1-\gamma}(\frac{1+\gamma}{2})\sqrt{\frac{\lambda T}{1-\gamma}} + \frac{4}{(1-\gamma)^2}(1-\lambda)^{\sqrt{\frac{(1-\gamma)T}{2\lambda}}} + \frac{2\gamma^2}{(1-\gamma)^2}(6\lambda^2(E-1)^2 + \lambda(E-1))\kappa \\
& + \left(\frac{12\gamma^2\lambda}{(1-\gamma)^2}\sqrt{E-1} + \frac{2\gamma^2\sqrt{\lambda}}{(1-\gamma)^2}\right)\sqrt{\lambda(E-1)\log\frac{|\mathcal{S}||\mathcal{A}|KT}{\delta}} + \frac{2\gamma}{(1-\gamma)^2}\sqrt{\frac{1}{K}\lambda\log\frac{|S||A|TK}{\delta}} \\
\leq\ & \frac{1}{1-\gamma}\exp\left\{-\frac{1}{2}\sqrt{(1-\gamma)\lambda T}\right\} + \frac{4}{(1-\gamma)^2}\exp\left\{-\sqrt{(1-\gamma)\lambda T}\right\} \\
& + \frac{2\gamma^2}{(1-\gamma)^2}(6\lambda^2(E-1)^2 + \lambda(E-1))\kappa \\
& + \left(\frac{12\gamma^2\lambda}{(1-\gamma)^2}\sqrt{E-1} + \frac{2\gamma^2\sqrt{\lambda}}{(1-\gamma)^2}\right)\sqrt{\lambda(E-1)\log\frac{|\mathcal{S}||\mathcal{A}|KT}{\delta}} + \frac{2\gamma}{(1-\gamma)^2}\sqrt{\frac{1}{K}\lambda\log\frac{|S||A|TK}{\delta}} \\
\leq\ & \frac{4}{(1-\gamma)^2}\exp\left\{-\frac{1}{2}\sqrt{(1-\gamma)\lambda T}\right\} + \frac{2\gamma^2}{(1-\gamma)^2}(6\lambda^2(E-1)^2 + \lambda(E-1))\kappa \\
& + \left(\frac{12\gamma^2\lambda}{(1-\gamma)^2}\sqrt{E-1} + \frac{2\gamma^2\sqrt{\lambda}}{(1-\gamma)^2}\right)\sqrt{\lambda(E-1)\log\frac{|\mathcal{S}||\mathcal{A}|KT}{\delta}} + \frac{2\gamma}{(1-\gamma)^2}\sqrt{\frac{1}{K}\lambda\log\frac{|S||A|TK}{\delta}}.
\end{aligned}
$$

## F Proof of Theorem 2

Let $|\mathcal{A}| = 1$, in which case $Q$-function coincides with the $V$-function. According to Algorithm 1, when $(t+1) \bmod E \neq 0$, we have

$$Q_{t+1}^k = \left((1-\lambda)I + \lambda\gamma P^k\right) Q_t^k + \lambda R.$$

Define $A^k \triangleq (1-\lambda)I + \lambda\gamma P^k$. We obtain the following recursion between two synchronization rounds:

$$Q_{(r+1)E}^k = (A^k)^E Q_{rE}^k + \left((A^k)^0 + \dots (A^k)^{E-1}\right) \lambda R.$$

Define

$$\bar{A}^{(\ell)} \triangleq \frac{1}{K} \sum_{k=1}^K (A^k)^\ell. \tag{27}$$

Note that $Q^*$ is the fixed point under the transition kernel $\bar{P}$, we have $\lambda R = \lambda(I - \gamma\bar{P})Q^* = (I - \bar{A}^{(1)})Q^*$ since $\bar{A}^{(1)} = I - \lambda(I - \gamma\bar{P})$. Furthermore, since $Q_{tE}^1, \dots, Q_{tE}^K$ are identical due to synchronization, we get

$$\bar{Q}_{(r+1)E} = \bar{A}^{(E)}\bar{Q}_{rE} + \left(I + \bar{A}^{(1)} + \dots \bar{A}^{(E-1)}\right)\left(I - \bar{A}^{(1)}\right)Q^*.$$

Consequently,

$$\begin{aligned}
\Delta_{(r+1)E} &= Q^* - \bar{Q}_{(r+1)E} \\
&= \bar{A}^{(E)}\Delta_{rE} + \left(\left(I - \bar{A}^{(E)}\right) - \left(I + \bar{A}^{(1)} + \dots \bar{A}^{(E-1)}\right)\left(I - \bar{A}^{(1)}\right)\right)Q^*. \tag{28}
\end{aligned}$$

Next, consider $|\mathcal{S}| = 2$ and even $K$ with

$$P^{2k-1} = \begin{bmatrix} 1 & 0 \\ 0 & 1 \end{bmatrix}, \quad P^{2k} = \begin{bmatrix} 0 & 1 \\ 1 & 0 \end{bmatrix}, \quad \text{for } k \in \mathbb{N}.$$

Then $\bar{P} = \frac{1}{2}\mathbf{1}\mathbf{1}^\top$, where $\mathbf{1}$ denotes the all ones vector. For the above transition kernels, we have

$$\frac{1}{k} \sum_{k=1}^K (P^k)^\ell = \begin{cases} I, & \ell \text{ even,} \\ \bar{P}, & \ell \text{ odd.} \end{cases}$$

Applying the definition of $\bar{A}^{(\ell)}$ in equation 27 yields that

$$\begin{aligned}
\bar{A}^{(\ell)} &= \frac{1}{K} \sum_{k=1}^K (A^k)^\ell \\
&= \frac{1}{K} \sum_{k=1}^K ((1-\lambda)I + \lambda\gamma P^k)^\ell \\
&= \frac{1}{K} \sum_{k=1}^K \sum_{j=0}^\ell \binom{\ell}{j}(\lambda\gamma P^k)^j((1-\lambda)I)^{\ell-j} \\
&= \sum_{j \text{ even}} \binom{\ell}{j}(1-\lambda)^{\ell-j}(\lambda\gamma)^j(I - \bar{P} + \bar{P}) + \sum_{j \text{ odd}} \binom{\ell}{j}(1-\lambda)^{\ell-j}(\lambda\gamma)^j\bar{P} \\
&= \underbrace{\frac{1}{2}((1-\lambda-\lambda\gamma)^\ell + (1-\lambda+\lambda\gamma)^\ell)}_{\triangleq\alpha_\ell}(I - \bar{P}) + \underbrace{(1-\lambda+\lambda\gamma)^\ell}_{\triangleq\beta_\ell}\bar{P} \\
&= \alpha_\ell(I - \bar{P}) + \beta_\ell\bar{P},
\end{aligned}$$

which is the eigen-decomposition of $\bar{A}^{(\ell)}$. Let

$$\lambda_1 \triangleq (1+\gamma)\lambda, \lambda_2 \triangleq (1-\gamma)\lambda, \quad \nu_1 = 1 - \lambda_1, \nu_2 = 1 - \lambda_2.$$

Then

$$\alpha_\ell = \frac{1}{2}(\nu_1^\ell + \nu_2^\ell), \quad \beta_\ell = \nu_2^\ell. \tag{29}$$

Note that $0 \leq \alpha \leq \beta \leq 1$ and $I - \bar{P}$ and $\bar{P}$ are orthogonal projection matrices satisfying $(I - \bar{P})\bar{P} = 0$. The matrices for the second term of the error on the right-hand side of 28 reduce to

$$\left(I + \bar{A}^{(1)} + \ldots \bar{A}^{(E-1)}\right)\left(I - \bar{A}^{(1)}\right)$$

$$= \left(\sum_{\ell=0}^{E-1} \alpha_\ell (I - \bar{P}) + \sum_{\ell=0}^{E-1} \beta_\ell \bar{P}\right)\left((\alpha_0 - \alpha_1)(I - \bar{P}) + (\beta_0 - \beta_1)\bar{P}\right)$$

$$= \left((1 - \alpha_1)\sum_{\ell=0}^{E-1} \alpha_\ell (I - \bar{P})^2 + (1 - \beta_1)\sum_{\ell=0}^{E-1} \beta_\ell \bar{P}^2\right) \text{ since } \alpha_0 = \beta_0 = 1$$

$$= \left((1 - \alpha_1)\sum_{\ell=0}^{E-1} \alpha_\ell (I - \bar{P}) + (1 - \beta_1)\sum_{\ell=0}^{E-1} \beta_\ell \bar{P}\right) \text{ since } (I - \bar{P}) \text{ and } \bar{P} \text{ are idempotent.}$$

It follow that

$$\left(I - \bar{A}^{(E)}\right) - \left(I + \bar{A}^{(1)} + \ldots \bar{A}^{(E-1)}\right)\left(I - \bar{A}^{(1)}\right)$$

$$= \underbrace{\left((1 - \alpha_E) - (1 - \alpha_1)\left(\sum_{i=0}^{E-1} \alpha_i\right)\right)}_{\triangleq \kappa_E}(I - \bar{P}) + \underbrace{\left((1 - \beta_E) - (1 - \beta_1)\left(\sum_{i=0}^{E-1} \beta_i\right)\right)}_{=0}\bar{P}.$$

Applying equation 29 yields that

$$\kappa_E = -\frac{\gamma}{2}\left(\frac{1 - \nu_2^E}{1 - \gamma} - \frac{1 - \nu_1^E}{1 + \gamma}\right). \tag{30}$$

It follows from equation 28 that the error evolves as

$$\Delta_{(r+1)E} = \left(\alpha_E(I - \bar{P}) + \beta_E \bar{P}\right)\Delta_{rE} + \kappa_E(I - \bar{P})Q^*,$$

which further yields the following full recursion of the error:

$$\Delta_{rE} = \left(\alpha_E(I - \bar{P}) + \beta_E \bar{P}\right)^r \Delta_0 + \sum_{\ell=0}^{r-1}\left(\alpha_E(I - \bar{P}) + \beta_E \bar{P}\right)^\ell \kappa_E(I - \bar{P})Q^*$$

$$= \left(\alpha_E^r(I - \bar{P}) + \beta_E^r \bar{P}\right)\Delta_0 + \sum_{\ell=0}^{r-1}\left(\alpha_E^\ell(I - \bar{P}) + \beta_E^\ell \bar{P}\right)\kappa_E(I - \bar{P})Q^*$$

$$\text{since } \left(\alpha_E(I - \bar{P}) + \beta_E \bar{P}\right)^\ell = \alpha_E^\ell(I - \bar{P}) + \beta_E^\ell \bar{P}, \forall \ell \in \mathbb{N}$$

$$= \left(\alpha_E^r(I - \bar{P}) + \beta_E^r \bar{P}\right)\Delta_0 + \frac{1 - \alpha_E^r}{1 - \alpha_E}\kappa_E(I - \bar{P})Q^*$$

$$= \left(\alpha_E^r + \frac{1 - \alpha_E^r}{1 - \alpha_E}\kappa_E\right)(I - \bar{P})Q^* + \beta_E^r \bar{P}Q^*,$$

where the last equality applied the zero initialization condition.

Note that $(I - \bar{P})Q^*$ and $\bar{P}Q^*$ are orthogonal vectors. Since $|\mathcal{S}| = 2$, we have

$$\|\Delta_{rE}\|_\infty \geq \frac{1}{\sqrt{2}}\|\Delta_{rE}\|_2 \geq \frac{\min\{\|(I - \bar{P})Q^*\|_2, \|\bar{P}Q^*\|_2\}}{\sqrt{2}} \cdot \max\left\{\left|\alpha_E^r + \frac{1 - \alpha_E^r}{1 - \alpha_E}\kappa_E\right|, \beta_E^r\right\}.$$

Since $Q^* = (I - \gamma\bar{P})^{-1}R = (I - \bar{P})R + \frac{1}{1-\gamma}\bar{P}R$, we obtain that

$$\|(I - \bar{P})Q^*\|_2 = \|(I - \bar{P})R\|_2, \qquad \|\bar{P}Q^*\|_2 = \frac{1}{1-\gamma}\|\bar{P}R\|_2.$$

Therefore, for the reward $R$ in general position, we have $\min\{\|(I-\bar{P})Q^*\|_2, \|\bar{P}Q^*\|_2\} \geq c_R$ for some constant $c_R$ depending on the reward function. It remain to analyze the coefficients as functions of $\lambda$. To this end, we introduce the following lemma:

**Lemma 5.** *The following properties hold:*

1. *Negativity: $\kappa_E < 0$;*

2. *Monotonicity: $\frac{\kappa_E}{1-\alpha_E}$ is monotonically decreasing for $\lambda \in (0, \frac{1}{1+\gamma})$;*

3. *Upper bound: $|\frac{\kappa_E}{1-\alpha_E}| \leq \frac{\gamma^2}{1-\gamma^2}$ for $\lambda \in (0, \frac{1}{1+\gamma})$;*

4. *Lower bound: if $(1+\gamma)\lambda \leq \frac{1}{2E}$, then $|\frac{\kappa_E}{1-\alpha_E}| \geq \frac{\lambda\gamma^2(E-1)}{4}$.*

*Proof.* We prove the properties separately.

1. Note that $\nu_1 < \nu_2$, $1 - \nu_1 = (1+\gamma)\lambda$, and $1 - \nu_2 = (1-\gamma)\lambda$. Then it follows from equation 30 that

$$\kappa_E = -\frac{\lambda\gamma}{2}\sum_{i=1}^{E-1}(\nu_2^i - \nu_1^i) < 0.$$

2. For the monotonicity, it suffices to show that $\frac{d}{d\lambda}\frac{\kappa_E}{1-\alpha_E} \leq 0$. We calculate the derivative as

$$\frac{d}{d\lambda}\frac{\kappa_E}{1-\alpha_E} = \frac{\gamma E(1-\nu_1^E)(1-\nu_2^E)}{2(1-\gamma^2)(1-\alpha_E)^2}\left(\frac{(1+\gamma)\nu_1^{E-1}}{1-\nu_1^E} - \frac{(1-\gamma)\nu_2^{E-1}}{1-\nu_2^E}\right).$$

   Note that

$$\frac{(1+\gamma)\nu_1^{E-1}}{1-\nu_1^E} - \frac{(1-\gamma)\nu_2^{E-1}}{1-\nu_2^E} = \frac{1}{\lambda}\left(\frac{\nu_1^{E-1}}{1+\nu_1+\cdots+\nu_1^{E-1}} - \frac{\nu_2^{E-1}}{1+\nu_2+\cdots+\nu_2^{E-1}}\right) \leq 0.$$

3. For the upper bound, it suffices to show the result at $\lambda = \frac{1}{1+\gamma}$ due to the negativity and monotonicity. At $\lambda = \frac{1}{1+\gamma}$, we have

$$\left|\frac{\kappa_E}{1-\alpha_E}\right| = \frac{\gamma}{1-\gamma^2}\left(\gamma - \frac{(\frac{2\gamma}{1+\gamma})^E}{2-(\frac{2\gamma}{1+\gamma})^E}\right) \leq \frac{\gamma^2}{1-\gamma^2}.$$

4. For the lower bound, the case $E = 1$ trivially holds. Next, consider $E \geq 2$. We have

$$\frac{\kappa_E}{1-\alpha_E} = -\frac{\gamma}{1-\gamma^2}\frac{(1+\gamma)(1-\nu_2^E) - (1-\gamma)(1-\nu_1^E)}{(1-\nu_1^E) + (1-\nu_2^E)}$$
$$= -\lambda\gamma\frac{\sum_{\ell=1}^{E-1}(\nu_2^\ell - \nu_1^\ell)}{(1-\nu_1^E) + (1-\nu_2^E)}.$$

   Note that $1 - nx \leq (1-x)^n \leq 1 - \frac{1}{2}nx$ for $n \geq 1$ and $0 \leq x \leq \frac{1}{n}$. Then, for $(1+\gamma)\lambda \leq \frac{1}{2E}$, we have

$$\nu_1^E = (1 - (1+\gamma)\lambda)^E \geq 1 - (1+\gamma)\lambda E \geq \frac{1}{2},$$
$$\nu_2^E = (1 - (1-\gamma)\lambda)^E \geq 1 - (1-\gamma)\lambda E.$$

Moreover, for all $x \in [\nu_1, \nu_2] \subseteq [0,1]$ and $\ell - 1 \le E$, we have

$$x^{\ell-1} \ge x^E \ge \nu_1^E \ge \frac{1}{2}.$$

We obtain that

$$\frac{\sum_{\ell=1}^{E-1}(\nu_2^\ell - \nu_1^\ell)}{(1 - \nu_1^E) + (1 - \nu_2^E)} \ge \frac{\sum_{\ell=1}^{E-1} \int_{\nu_1}^{\nu_2} \ell \cdot x^{\ell-1} dx}{2\lambda E} \ge \frac{\sum_{\ell=1}^{E-1} \ell \frac{1}{2}(\nu_2 - \nu_1)}{2\lambda E} = \frac{1}{4}\gamma(E-1).$$

The proof is completed. $\qquad\square$

We consider two regimes of the stepsize separated by $\lambda_0 \triangleq \frac{\log r}{(1-\gamma)rE} < \frac{1}{1+\gamma}$, where the dominating error is due to the small stepsize and the environment heterogeneity, respectively:

**Slow rate due to small stepsize when $\lambda \le \lambda_0$.** Since $\beta_E^r$ monotonically decreases as $\lambda$ increases,

$$\beta_E^r = (1 - (1-\gamma)\lambda)^{rE} \ge (1 - (1-\gamma)\lambda_0)^{rE} = \left(1 - \frac{\log t}{rE}\right)^{rE}.$$

Note that $\frac{\log r}{rE} \in (0, \frac{1}{2})$, applying the fact $\log(1-x) + x \ge -x^2$ for $x \in [0, \frac{1}{2}]$ yields that

$$\log\left(1 - \frac{\log r}{rE}\right) + \frac{\log r}{rE} \ge -\left(\frac{\log r}{rE}\right)^2 \ge -\frac{1}{rE}.$$

Then we get

$$\beta_E^r \ge \left(1 - \frac{\log r}{rE}\right)^{rE} \ge \frac{1}{er}.$$

**Slow rate due to environment heterogeneity when $\lambda \ge \lambda_0$.** Recall that $\lambda < \frac{1}{1+\gamma}$. Applying the triangle inequality yields that

$$\left| \alpha_E^r + \frac{1 - \alpha_E^r}{1 - \alpha_E} \kappa_E \right| \ge \left| \frac{\kappa_E}{1 - \alpha_E} \right| - \left(1 + \left| \frac{\kappa_E}{1 - \alpha_E} \right|\right) \alpha_E^r.$$

For the first term, by the negativety and monotonicity in Lemma 5, it suffices to show the lower bound at $\lambda = \lambda_0$. Since $\lambda < \frac{1}{1+\gamma}$, then $\alpha_E = \frac{1}{2}\left((1 - (1-\gamma)\lambda)^E + (1 - (1+\gamma)\lambda)^E\right)$ decreases as $\lambda$ increases. For $t \gtrsim \frac{1}{1-\gamma} \log \frac{1}{1-\gamma}$ such that $(1+\gamma)\lambda_0 \le \frac{1}{2E}$, we apply the lower bound in Lemma 5 and obtain that

$$\left| \frac{\kappa_E}{1 - \alpha_E} \right| \gtrsim \gamma^2 \frac{\log r}{(1-\gamma)r}.$$

Additionally, applying the upper bound in Lemma 5 yields

$$\left(1 + \left| \frac{\kappa_E}{1 - \alpha_E} \right|\right) \alpha_E^r \le \frac{\nu_2^{rE}}{1 - \gamma^2} = \frac{(1 - (1-\gamma)\lambda)^{rE}}{1 - \gamma^2} \le \frac{1}{(1-\gamma^2)r}.$$

The conclusion follows.

# G    Additional experiments

## G.1    Impacts of $E$ on homogeneous settings.

For the homogeneous settings, in addition to $E = 10$, we also consider $E = \{1, 20, 40, \infty\}$, where $E = \infty$ means no communication among the agents throughout the entire learning process. Similar to Figure 2b, there is no obvious two-phase phenomenon even in the extreme case when $E = \infty$. Also, though there is indeed performance degradation caused by larger $E$, the overall performance degradation is nearly negligible compared with the heterogeneous settings shown in Figures 2a and 3.

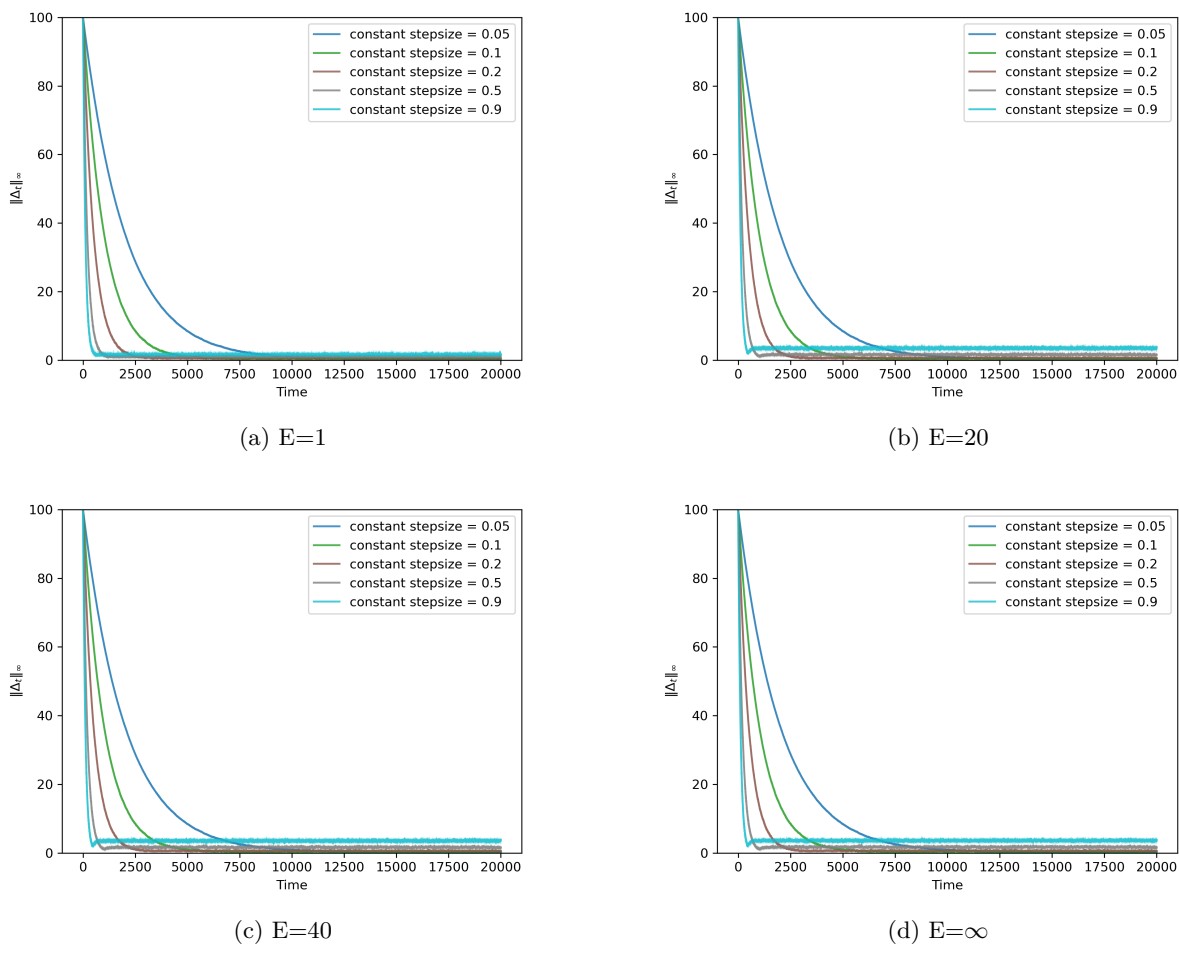

Figure 5: Homogeneous FQL with varying $E$.

## G.2    Different target error levels.

In Figure 6, we show the error levels that these training strategies can achieve within a time horizon $T = 20,000$. The tolerance levels are $10\%, 5\%, 3\%,$ and $1\%$ of the initial error $\|\Delta_0\|_\infty$, respectively. At a high level, choosing different stepsizes for phases 1 and 2 can speed up convergence.

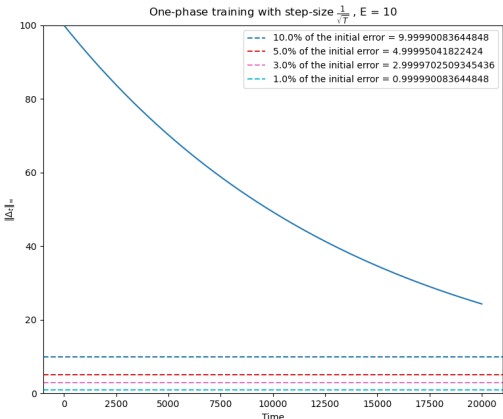

(a) One common $\lambda = \frac{1}{\sqrt{T}}$ throughout. $\|\Delta_t\|_\infty$ does meet any of the tolerance levels within 20000 iterations

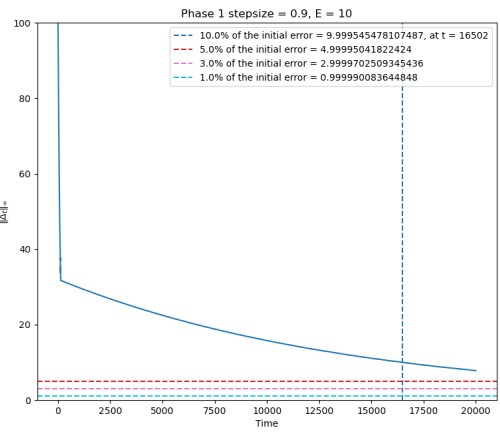

(b) With a phase 1 stepsize of 0.9, it meets the 10% tolerance level at iteration 16502.

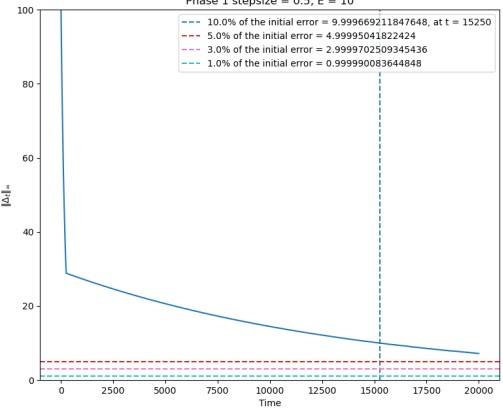

(c) With a phase 1 stepsize of 0.5, it meets the 10% tolerance level at iteration 15250.

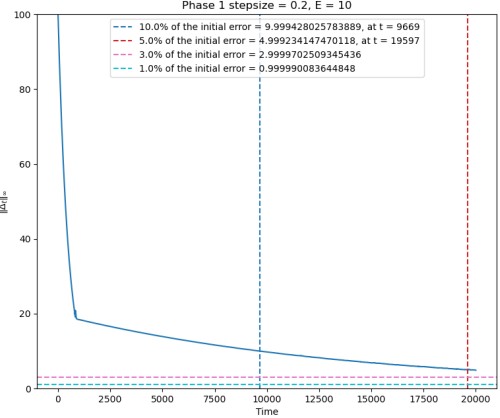

(d) With a phase 1 stepsize of 0.2, it meets the 10% and 5% tolerance level at iterations 9669 and 19597, respectively.

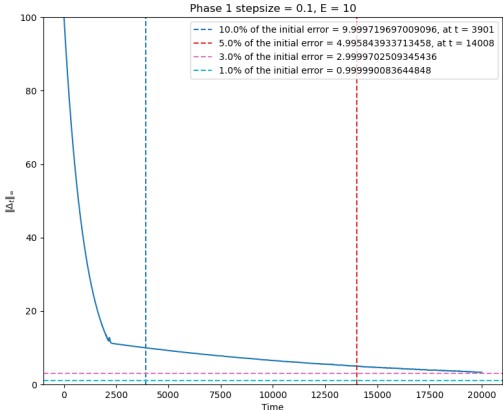

(e) With a phase 1 stepsize of 0.1, it meets the 10% and 5% tolerance level at iterations 3901 and 14008, respectively.

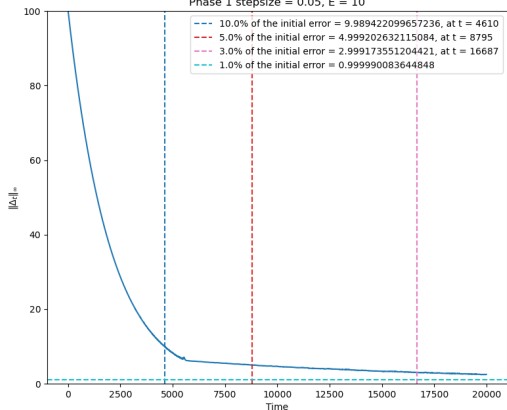

(f) With a phase 1 stepsize of 0.05, it meets the 10%, 5%, and 3% tolerance levels at iterations 4610, 8795, and 16687, respectively.

Figure 6: Convergence performance of different tolerance levels of different stepsize choices. The horizontal dashed lines represent the tolerance levels not met, while the vertical dashed lines indicate the iterations at which the training processes meet the corresponding tolerance levels.

