# OpenReview forum: "On the Convergence Rates of Federated Q-Learning across Heterogeneous Environments"
_TMLR — Rejected by TMLR_

### Review · Reviewer_Nzmh · 2024-07-13

**Summary Of Contributions:**

This paper focused on the synchronous federated Q-Learning algorithm under environmental heterogeneity and showed the performance degradation with regard to the synchronous period $E$, which was validated via the proof, the information bound, and the numerical experiment. In addition, the authors found the two-phase phenomenon (the error first decreasing to a minimum point and then bouncing back rapidly before stabilizing around some fixed values) in their numerical experiments. Overall, I am positive about the rating of this paper if the authors can address my concerns.

**Audience:**

Yes

**Broader Impact Concerns:**

Do not apply. This is a theoretical work.

**Claims And Evidence:**

Yes

**Requested Changes:**

I am positive about the technique soundness of this paper and only wish the authors to address the concerns in the weakness part. In addition, I wonder whether there are any mathematical explanations regarding the two-stage phenomenon.

**Strengths And Weaknesses:**

Strength:
1.	The authors provided solid mathematical analysis for the error bound of the degradation and also showed a lower bound of $\Omega(E/T)$.
2.	The authors provided comprehensive numerical experiments that show the degradation as well as the two-phase phenomenon, which is quite surprising.


Weakness:
1.	The authors didn’t discuss the communication cost of their federated algorithm. In fact, even under the degradation, Corollary 1 indicates that their algorithm can reach a better communication cost compared to $O(T)$.

2.	When discussing degradation, the authors only limit their discussion to fixed step sizes. It is still unclear about whether the degradation can be alleviated when we use diminishing step sizes like [1] or [2]. It would be better if the authors could provide some reasonings regarding the situations for diminishing step sizes.

3.	I suggest the authors provide a more rigorous statement in Theorem 2. First, the RHS in the display mode equation does not include terms related to $O(1/\sqrt{T})$, which seems wrong if we adopt the form $= \Omega(something)$ instead of $\geq \Omega(something)$. Second, $\Omega(E/T)$ seems to hide some constant factors. If the authors are certain that it only hides numerical constants, please state it clearly.

4. There are typos and undefined notations throughout the whole paper. I suggest the authors conduct additional rounds of proofreading. The following content provides an incomprehensive list.

(a)	In “contributions” in section 1, $K$ is mentioned before the explanation.

(b)	There might be ambiguity in the function “mod” if it is not clearly undefined.

(c)	In Lemma 3, $\mbox{max} _{s,a} \|\|\cdot\|\| _\infty$ is wrong. The same issue also appears in the last display mode equation in Appendix C.

(d)	In the second display mode equation in section 5.1, $\lambda _t$ is wrong.

(e)	In the third line of Appendix B, $t+1\ \mbox{mod}\ \neq 0$ is wrong.

(f)	In the second line after the first display mode equation in Appendix B, $\|\| _1$ is wrong.

(g)	In Appendix C, $V _t,Q _t$ are undefined.

(h)	In the line after equation 18, $\delta\in\delta$ is wrong.

(i)	When using Hoeffding’s inequality in equation 19, more explanation will be better.

(j)	In equation 21, $\chi(t)$ is wrong.

(k)	In equation 19 and equation 20, $|\cdot|$ is wrong.

(l)	Between equation 22 and equation 23, $X(\cdot))$ is wrong.

(m)	In the line after equation 25, please add equation references for “1,2,3”.

(n)	At the beginning of page 22, please put some effort into handling the situation that $\beta,L$ might not be integers.

[1] Is Q-learning Provably Efficient?
[2] Federated Q-learning: linear regret speedup with low communication cost

---

> ### Author Response · Authors · 2024-07-25
> **A more rigorous statement in Theorem 2.**
>
> Thank you for pointing out the problem. We will change $=\Omega(\frac{E}{T})$ to $\geq\Omega(\frac{E}{T})$.
>
> Moreover, at the end of page 26, we have two hidden constants, the first one occurs when we need a lower bound of $r$ to make sure $  (1+\gamma)\lambda_0<\frac{1}{2E}$. This can be extended to an exact solution by introducing Lambert W function.
>
> For $(1+\gamma)\lambda_0\le \frac{1}{2E}$, it is equivalent to $\frac{\log r}{r} \leq \frac{1-\gamma}{2(1+\gamma)}$. Since $\frac{\log r}{r}$ monotonically decreases when $r>e$, it is sufficient to find $\frac{\log r}{r} = \frac{1-\gamma}{2(1+\gamma)}$, and let $r$ be greater than the solution. That is,
> $$\frac{\log (r)}{e^{\log(r)}} = \frac{1-\gamma}{2(1+\gamma)}\implies -\frac{\log (r)}{e^{\log(r)}}=- \frac{1-\gamma}{2(1+\gamma)} \implies -\log(r) e^{-\log(r)}=- \frac{1-\gamma}{2(1+\gamma)}\implies r = \exp\left\\{-W_{-1}\left(-\frac{1-\gamma}{2(1+\gamma)}\right)\right\\}, \text{ where } W_{-1} \text{ is the Lambert $W$ function}$$
> Therefore, for $r\geq \exp\left\\{-W_{-1}\left(-\frac{1-\gamma}{2(1+\gamma)}\right)\right\\}$, we have $(1+\gamma)\lambda_0\le \frac{1}{2E}$.
>
> The second hidden constant occurs when giving the lower bound of $\left|\frac{\kappa_E}{1-\alpha_E}\right|$. A precise expression is derived in the following lines:
> Applying the lower bound in Lemma 5,
> $$\left|\frac{\kappa_E}{1-\alpha_E}\right| \geq \frac{\lambda_0 \gamma^2(E-1)}{4}\geq \frac{\frac{\log r}{(1-\gamma) r E} \gamma^2(E-1)}{4}  \geq \frac{(E-1)}{4E} \gamma^2 \frac{\log r}{(1-\gamma)r}$$
>
> In addition, the effect of this precise bound propagates to the final lower bound.
>
> $$\left|{\alpha_E^r + \frac{1-\alpha_E^r}{1-\alpha_E}\kappa_E }\right|
> \ge \left|{\frac{\kappa_E}{1-\alpha_E}}\right| - \left({1+ \left|{\frac{\kappa_E}{1-\alpha_E}}\right| }\right)\alpha_E^r \geq \frac{(E-1)}{4E} \gamma^2 \frac{\log r}{(1-\gamma)r} -  \frac{1}{(1-\gamma^2)r}  = \frac{1}{(1-\gamma^2)r} \left({(1+\gamma)\gamma^2\log(r)(E-1)/(4E) -1}\right)$$
> $$ = \frac{1}{(1-\gamma)r} \left({\frac{\gamma^2\log(r)(E-1)-4E/(1+\gamma)}{4E}}\right)$$
>
> Therefore,
> $$   \left|{\Delta_{rE}}\right|
> \ge \frac{\min\\{\|(I-\bar P)Q^*\|_2, \|\bar P Q^*\|_2 \\}}{\sqrt{2}} \cdot \max\left\\{|\alpha_E^r + \frac{1-\alpha_E^r}{1-\alpha_E}\kappa_E|, \beta_E^r\right\\} \ge \frac{c_R}{\sqrt{2}}\max\left\\{|\alpha_E^r + \frac{1-\alpha_E^r}{1-\alpha_E}\kappa_E|, \beta_E^r\right\\}  \geq \frac{c_R}{\sqrt{2}}\max\left\\{\frac{1}{(1-\gamma)r} \left(\frac{\gamma^2\log(r)(E-1)-4E/(1+\gamma)}{4E}\right), \frac{1}{er}\right\\}$$
> $$ = \frac{c_R}{\sqrt{2}}\max\left\\{\frac{E}{(1-\gamma)T} \left(\frac{\gamma^2\log(r)(E-1)-4E/(1+\gamma)}{4E}\right), \frac{E}{eT}\right\\}$$
>
> We can choose $\log(r)\geq \frac{4E(\gamma+2)}{(1+\gamma)\gamma^2(E-1)}, E\geq 2$ so that $\left(\frac{\gamma^2\log(r)(E-1)-4E/(1+\gamma)}{4E}\right)\geq 1$. Then the first term inside the max operator is bigger, and $c_R$ is also a numeric constant when $R$ is given (as described in our original paper). Therefore, the precise lower bound is $$\|\Delta_{rE}\|_{\infty} \geq \frac{c_R}{\sqrt{2}}\frac{E}{(1-\gamma)T} \left(\frac{\gamma^2\log(r)(E-1)-4E/(1+\gamma)}{4E}\right) \geq \frac{c_R}{\sqrt{2}}\frac{E}{(1-\gamma)T} $$

---

> ### Author Response · Authors · 2024-07-31
> **On the other issues**
>
> **Weakness 1:** Thank you so much for your valuable and constructive comments!
> Yes, we will add a remark talking about the communication cost.
>
> **Weakness 2:** We ran extra experiments with diminishing stepsizes to compare different synchronous period $E$. It turns out the convergence behavior also depends on how fast the stepsize diminishes. Specifically, with $\lambda_i = \frac{1}{\sqrt{i+1}} \text{ or } \frac{1}{(i+1)^{0.4}} \text{ or } \frac{1}{(i+1)^{0.3}}$ (stepsize decays relatively slow), the degradation is still very obvious. However, for $\lambda_i = \frac{1}{i+1} \text{ or } \frac{1}{(i+1)^{0.7}} \text{ or } \frac{c+1}{i+1}$(stepsize decays relatively fast), we observe that the overall convergence is very slow and the effect of $E$ is not significant. If needed, we can investigate more and add more discussion in this direction.
>
> **On the mathematical explanation of the two-phase phenomenon:**
> In our lower-bound proof, we have a recurrence relation in the following form:
>
> $$\Delta_{rE} = \left({\alpha_E^r + \frac{1-\alpha_E^r}{1-\alpha_E}\kappa_E  }\right) (I-\bar P)Q^* + \beta_E^r \bar P Q^*$$
> Note that $(I-\bar P)Q^*$ and $\bar P Q^*$ are orthogonal to each other and their magnitude are independent of $r$. Then what really controls this term is the coefficients. The second coefficient $\beta_E^r$ is well-behaved since it is between $0$ and $1$, and decreases as $r$ increases. However, if the stepsize is not well-controlled, the first coefficient $\left({\alpha_E^r + \frac{1-\alpha_E^r}{1-\alpha_E}\kappa_E  }\right)$(which relates to heterogeneity) goes from positive to negative and continues to decrease until converging to some negative real number. Therefore, its absolute value decreases first and then increase, causing the two-phase phenomenon.

---

### Review · Reviewer_tNAP · 2024-07-15

**Summary Of Contributions:**

This paper investigates federated Q-learning with environment heterogeneity, where agents interact with local MDPs having different transition kernels. The study includes a finite-time convergence analysis of Q-learning under FL settings using FedAvg. The convergence bound explores the impact of synchronization period (E) and the degree of heterogeneity ($\kappa$) on convergence speed. Additionally, the paper identifies a significant slowdown in federated Q-learning due to environment heterogeneity and large synchronization periods (E), supported by convergence analysis and experimental results, and suggested two-phase training using different learning rates, alleviating the slowdown.

**Audience:**

Yes

**Broader Impact Concerns:**

Broader impact is properly addressed

**Claims And Evidence:**

Yes

**Requested Changes:**

1. To address W1, the paper should include detailed comparisons with the convergence bounds and implications presented in previous works on FRL involving environment heterogeneity.
2. To address W2, the paper should discuss unique observations about convergence slowdown with less communications discovered in their specific settings that have not been covered in previous FRL and FL literature.
3. To address W3, the paper should provide comparisons demonstrating the advantages of the proposed two-phase training method over other FL and RL approaches designed to manage client drifts resulting from extended synchronization periods.

**Strengths And Weaknesses:**

Strengths
1. They conducted a finite-time convergence analysis of federated Q-learning in heterogeneous environments.
2. They extensively discussed the slowdown in convergence caused by extended synchronization periods in heterogeneous environments and empirically demonstrated the effectiveness of switching learning rates in such scenarios.

Weaknesses
Overall, the paper appears to lack comparisons with other related works in terms of both theoretical and empirical analyses, and the contribution of this paper is somewhat unclear.
1. The convergence analysis and implications presented in the paper seems similar to that of previous works on federated RL involving heterogeneous environments [1,2]. These works have also demonstrated that convergence bounds increase with the level of heterogeneity ($\kappa$) and synchronization periods (E). Despite listing these works in the related section, it remains unclear what novel discoveries or improvements the analysis offers compared to previous studies.
2. Convergence slowdown due to large synchronization periods is a well-known issue in classic FL. In FL, as agents perform multiple local updates, client deviations can lead to convergence slowdown. However, it is unclear from this paper how the observed convergence slowdown differs from general FL challenges.
3. The effectiveness of the proposed two-phase training with different learning rates lacks clarity due to the absence of comparisons with other FL and FRL methods designed to manage client deviations arising from lengthy synchronization periods. For instance, it's unclear whether the proposed method outperforms commonly used techniques like linearly decreasing learning rates, widely adopted in various RL and FL scenarios.

[1] Federated temporal difference learning with linear function approximation under environmental heterogeneity, Wang et al., arXiv preprint arXiv:2302.02212, 2023.

[2] Finite-time analysis of on-policy heterogeneous federated reinforcement learning, Zhang et al., In The Twelfth International Conference on Learning Representations, 2023.

---

> ### Author Response · Authors · 2024-07-17
> **On the novelty compared with [1][2]**
>
> We respectfully disagree that our analysis shares similarities with [1] and [2]. We carefully re-examined the analysis of [1] and [2]. We believe this comment arises from the reviewer's misreading of our submission, and the two papers referred to (i.e., [1] and [2]).
>
> [1] and [2] studied a federated version of TD learning and SARSA, whereas our paper is on a federated version of Q-Learning.
> Neither the results nor analysis of [1] and [2] can be nontrivially extended to our setting. From a theoretical perspective, TD(0) methods only estimate the value function $V(s)$ while Q-Learning estimates the action-value function $Q(s,a)$, meaning that TD(0) is not appropriate for control problems because the value function does not directly give us how good an action is given a particular state. SARSA does estimate $Q(s,a)$, however, due to SARSA's update rule, it must sample an additional action $s'$ and use the $R(s,a)+\gamma Q(s',a')$ as the new estimate of the Q-function, while in Q-Learning, the new estimate is $R(s,a)+\gamma \max_{a'}Q(s',a')$, where "max" is a nonlinear mapping of the Q-fucntion. As for the applicational differences, TD(0), SARSA, and Q-Learning each have distinct use cases, underscoring the unique contribution of our work focusing on Q-Learning. TD(0) is primarily used for prediction problems, where the goal is to estimate the value function of a given policy without focusing on control or decision-making tasks. In contrast, SARSA, with its on-policy nature, is well-suited for real-time decision-making and control problems, such as robotics or online learning tasks, where the agent needs to continuously update its policy based on its interactions with the environment.
> Q-Learning, being an off-policy algorithm, is particularly effective for offline learning and planning tasks, such as game-playing or optimization problems, where the objective is to find the optimal policy that maximizes long-term rewards regardless of the agent's current behavior. Therefore, the significance of our work on Q-Learning should not be overlooked, as it addresses a critical and distinct aspect of reinforcement learning, focusing on optimal policy learning and planning, which are not adequately covered by TD(0) or SARSA.
>
> More importantly, both [1] and [2] characterized the upper bounds of finite-time convergence rates only. Hence, their results cannot support any implications on how fundamentally the convergence rates are impacted by the heterogeneity and synchronization period $E$. This is because upper bounds can be loose. Even if these upper bounds increase with the level of $\kappa$ and $E$, the convergence rates themselves may not slow down. In contrast, we derived a lower bound on the convergence rates, showing the fundamental limitation of multiple local updates (i.e., $E>1$) in the presence of environmental heterogeneity. To the best of our knowledge, this is the first result of its kind.
>
> Last but not least, in the main results of [1] (their Theorem 2), the error is upper bounded by a sum of four terms, with the last term containing the impacts of environmental heterogeneity. This upper bound suffers two limitations. First, the entire upper bound does not decay to zero as $T\to \infty$. In other words, they only show the error is eventually bounded rather than diminishes. Second, the impacts of environmental heterogeneity on the convergence speed are overlooked. Their upper bounds only imply the smaller the environmental heterogeneity, the smaller the final error bound is. The main convergence results in [2] face the same issues. In sharp contrast, in our results, when the stepsize decays in T (the total number of iterations), our derived error upper bounds approach 0 as $T\to \infty$.

---

> ### Author Response · Authors · 2024-07-17
> **On the novelty of convergence slow-down.**
>
> What we showed is that the slowdown occurs as long as $E>1$ in the presence of environmental heterogeneity. Yet, in traditional FL, the commonly agreed criticism on the use of large synchronization period $E$ only, which are mostly validated through experiments with extreme data heterogeneity.
>
> We also respectfully disagree with your comment that "In FL, as agents perform multiple local updates, client deviations can lead to convergence slowdown." This statement is not accurate.
> In [R.1] (the most classical FL paper), Figure 2 clearly shows that within a certain range of choices of E and batch sizes, larger $E$ leads to faster convergence in terms of communication rounds -- not clear slow-down in terms of total updates.
> In addition, in a non-parametric statistical learning setting, [R.2] mathematically showed that with multiple $E$ (up to a certain range) and properly chosen learning rates, the convergence speed (in terms of communication rounds) is faster when $E$ is larger.
> Overall, in general FL, there are multiple types of data heterogeneity with each of such heterogeneity coming in different levels.
> Whether and how multiple local updates will affect the convergence is not well-studied.
>
> [R.1] McMahan - Moore- Ramage- Hampson - Arcas, "Communication-Efficient Learning of Deep Networks from Decentralized Data", AISTATS 2017
>
> [R.2] Su - Xu - Yang, "A Non-parametric View of FedAvg and FedProx:
> Beyond Stationary Points", JMLR 2023
>
> [R.3] Kairouz et al. Advances and open problems in federated learning, Foundations and Trends® in Machine Learning, 2021.
> Section 3.1 in the following version for an overview of data heterogeneity
>
> https://www.andrew.cmu.edu/user/gaurij/Papers/kairouz2019advances.pdf

---

### Review · Reviewer_V1jm · 2024-07-15

**Summary Of Contributions:**

This paper studies the sample complexity (equivalently, iteration complexity) of synchronous federated Q-learning algorithm with environment heterogeneity. The theoretical contribution are two aspects. The first contribution is the finite-sample guarantee of the algorithm. The second contribution is the example of elaborating the phenomenon that the convergence rate can be linear dependent on local updates $E$. The empirical contribution is the numerical experiments on one setting. However, there are several issues that make me feel it is an inappropriate manuscript to be published at TMLR. The main concern to me is the contribution is not significant enough. Below are some comments.

**Audience:**

Yes

**Broader Impact Concerns:**

No.

**Claims And Evidence:**

Yes

**Requested Changes:**

1. For the first theoretical contribution, the only difference from \cite{woo2023blessing} is the additional environment heterogeneous assumption. In Corollary 1, the dominating term in Corollary 1 is of sample complexity $\widetilde{\mathcal{O}}\left(\frac{SA}{(1-\gamma)^6\varepsilon^2K}\right)$,
    which is independent with environment heterogeneity $\kappa$ and worse than the result in \cite{woo2023blessing} by a factor $(1-\gamma)^{-1}$. The authors should compare the results to elaborate whether this additional $(1-\gamma)^{-1}$ factor is due to the environment heterogeneity.

2. I would recommend the authors add the asynchronous case to enrich the results. The current results are not sufficient for a journal paper though it may be sufficient for a conference paper. Moreover, I would also recommend the authors add a few more experiments on other more complex environments.

3. In the experiments, Fig 3 does not really match the theoretical results of the theory. The theory tells us the algorithm converges but Fig 3 shows the algorithm may fail to converge with large local updates. The authors need to elaborate the phenomenon with more details.

Below I list some minor comments about this paper.
1. In the description of Algorithm 1, please elaborate how $s_t^k(s,a)$ is sampled. I guess it follows from $P^k(\cdot|s,a)$.
2. In the line 6 of Algorithm 1, it should be $s_t^k(s,a)$ instead of $s_t(s,a)$.
3. In Remark 1 last line, I guess it is $\lambda=\Omega(1/T)$ instead of $\lambda=\omega(1/T)$.
4. In Page 15, third line from the bottom, change $\widetilde{P}_t^k\||_1$ to $\||\widetilde{P}_t^k\||_1$.
5. In Section E, the definition of $X_{i,k}(s,a)=\frac{1}{K}\gamma\lambda(1-\lambda)^i(P^k-\widetilde{P}^k_{i-k})V^*$ is somehow misleading as the LHS is a scalar and RHS is a $SA$-vector. I suggest the authors define it as $X_{i,k}=\frac{1}{K}\gamma\lambda(1-\lambda)^i(P^k-\widetilde{P}^k_{i-k})V^*$.
6. In Fig 5, FQL is never defined in the paper, though I know it may mean Federated Q-learning.

**Strengths And Weaknesses:**

See requested changes.

---

> ### Author Response · Authors · 2024-07-16
> **Response: On the significance of our results.**
>
> We respectfully disagree that our contributions are not significant enough.
>
> First and foremost, the lower bound on the convergence rates in the presence of environmental heterogeneity is the first result of its kind, showing the fundamental limitation of multiple local updates. Constructing concrete challenging problem instances is a common way to derive lower bounds in many communities such as statistical learning, theoretical computer science, and distributed computing.
>
> We are also the first to observe that using time-invariant learning rates, the convergence of federated Q-learning exhibits a two-phase phenomenon.
>
> Both the lower bound and the identified two-phase phenomenon can have significant practical implications, suggesting either adaptive synchronization frequency $E$ or adaptive learning rates.

---

> ### Author Response · Authors · 2024-07-16
> **Response: On the sample complexity in terms of $(1-\gamma)$**
>
> Thank you very much for sharing your concern on this.
> Our Theorem 1 does not lead to an additional $(1-\gamma)^{-1}$ factor. Moreover, due to the environmental heterogeneity, the term with sample complexity $\tilde{O}(SA/(1-\gamma)^5 \epsilon^2 K)$ is not necessarily the dominating term.
>
> Choosing the learning rate $\lambda = \frac{4\log^2(TK)}{T(1-\gamma)}$, the four terms in the upper bound of Theorem 1 can be further bounded as
> $$
> \frac{1}{(1-\gamma)^2} \exp^{-\frac{1}{2}\sqrt{(1-\gamma)\lambda T}} \le \epsilon, ~~\text{when } T \gtrsim \frac{1}{K\epsilon (1-\gamma)^2}.
> $$
>
> $$
> \frac{\gamma^2}{(1-\gamma)^2} (\lambda^2(E-1)^2+\lambda(E-1)) \kappa \le \epsilon, ~~\text{when } T \gtrsim \max \left\lbrace\frac{(E-1)\kappa}{(1-\gamma)^3\epsilon \log^2(TK)}, ~ \frac{(E-1)\sqrt{\kappa}}{(1-\gamma)^2 \sqrt{\epsilon}\log^2(TK)} \right\rbrace.
> $$
>
> $$
> (\frac{\gamma^2 \lambda}{(1-\gamma)^2}\sqrt{E-1}+\frac{\gamma^2 \sqrt{\lambda}}{(1-\gamma)^2})\sqrt{\lambda (E-1) \log \frac{|\mathcal{S}||\mathcal{A}|KT}{\delta}}\le \epsilon, ~~\text{when } T \gtrsim \max \left\lbrace\frac{\sqrt{E-1}\sqrt{\log \frac{|\mathcal{S}||\mathcal{A}|KT}{\delta}}}{(1-\gamma)^3 \log^2(TK) \epsilon},  \frac{(E-1)^{2/3} \log^{1/3} \frac{|\mathcal{S}||\mathcal{A}|KT}{\delta}}{(1-\gamma)^{7/3}\log^2(TK) \epsilon^{2/3}}\right\rbrace,
> $$
>
> $$
> \frac{\gamma}{(1-\gamma)^2}\sqrt{\frac{1}{K}\lambda \log\frac{|S||A|TK}{\delta}} \le \epsilon,  ~~\text{when } T \gtrsim \frac{\log \frac{|\mathcal{S}||\mathcal{A}|KT}{\delta}\log^2(TK)}{K\epsilon^2(1-\gamma)^5}.
> $$
> Depending on the values of $K$ and $E$, the term that involves environmental heterogeneity may be the dominating term.
> Fundamentally, as illustrated in Fig.2, the randomness in the sampling does not dominate the convergence.
>
>
> Yet, we do agree that the sample complexity derived directly from Corollary 1 could be loose.
> We had the current form of Corollary 1 to highlight the convergence rate in terms of $(E-1)/T \times \kappa$. If given the chance, we will add the discussion in terms of $(1-\gamma)$ in our revision.

---

> ### Author Response · Authors · 2024-07-16
> **Response: On the alignment of theory and Fig.\,3**
>
> We believe this is a misreading of our results.
>
> Based on our Theorem 1, the key to guaranteeing convergence is to carefully choose the step size $\lambda$ in terms of $T$. If we use a constant $\lambda$ that is independent of $T$, such as the values we tested and plotted in Figure 3 (0.9, 0.5, 0.2, 0.1, 0.05), the upper bound in Theorem 1 shows that only the first term decays as $T$ increases, while the remaining terms do not. Consequently, convergence is not guaranteed with these constant values. However, if we choose $\lambda = \frac{1}{\sqrt{T}}$, the algorithm is guaranteed to converge, as demonstrated in Figure 4 (light blue curve).
>
> In summary, our theoretical results are consistent with the experimental findings.

---

### Decision · Action_Editor_ipNf · 2024-09-25

**Recommendation:** Reject

**Comment:**

This submission had additional private discussions between myself and the three reviewers about their decision recommendations after the rebuttal.

The consensus is the following: The reviewers agree that the submission has potential, however they all have concerns with respect to the TMLR acceptance criterion: "Are the claims made in the submission supported by accurate, convincing and clear evidence?". More concrete suggestions are given in "Claims and Evidence" section.

**Audience:**

The reviewing team agrees that the submission satisfies this criterion.

**Claims And Evidence:**

This submission had additional private discussions between myself and the three reviewers about their decision recommendations after the rebuttal. The reviewing team decided that the submission unfortunately does not satisfy this criterion in the current form. In particular, reviewers raised concerns about the following concrete points:

- The differences between the submission and the existing work such as the work of Woo et al., 2023 are not highlighted sufficiently and the presentation of the results and the analyses was found to be unclear to the reviewing team, leading to concerns about the "clarity" of the evidence. For example, Reviewer V1jm raised some important concerns about the complexity in Theorem 1, Corollary 1. In particular, in the submission and the rebuttal, there are confusions about the dominating term as well as the comparisons of the order in dominating terms with the prior work. The rebuttal has a long calculation for bounding the terms, but the reviewing team thinks the discussion is still unclear. The conditions for $T$ and other terms as well as complexity need to be explicitly derived (with precise constants) and comparisons with relevant work of Woo et al, 2023 should be clearly highlighted. This will include both the differences in the setting and how it affects the complexity. The authors have some part of it in their rebuttal but it needs to be made more explicit as the discussion is not clear to the reviewing team at the moment.

- Another issue about clarity was pointed out by Reviewer Nzmh (also acknowledged by the authors) regarding the constants in Theorem 2. The constants should be written explicitly and accurately. The discussion in the rebuttal is not found to be clear to the reviewing team. The reviewers believe that an additional review period would be needed to ensure that the claims are accurate.

- Reviewer V1jm and Reviewer Nzmh also pointed out inaccuracies in the proofs which they suspect may affect the accuracy of the results.

In summary, the reviewing team acknowledges that the claimed results can be interesting to the community, however the "Claims and Evidence" aspect should be improved in terms of clarity, accuracy and therefore in terms of being convincing. Before further consideration for publication, the concerns of the reviewers should be addressed in a major revision.

**Resubmission Of Major Revision:**

The authors may consider submitting a major revision at a later time.